# Hierarchized phosphotarget binding by the seven human 14-3-3 isoforms

Gergo Gogl [1,3 ✉], Kristina V. Tugaeva[2,3], Pascal Eberling[1], Camille Kostmann[1], Gilles Trave [1 ✉] & Nikolai N. Sluchanko [2 ✉]

The seven 14-3-3 isoforms are highly abundant human proteins encoded by similar yet distinct genes. 14-3-3 proteins recognize phosphorylated motifs within numerous human and viral proteins. Here, we analyze by X-ray crystallography, fluorescence polarization, mutagenesis and fusicoccin-mediated modulation the structural basis and druggability of 14-3-3 binding to four E6 oncoproteins of tumorigenic human papillomaviruses. 14-3-3 isoforms bind variant and mutated phospho-motifs of E6 and unrelated protein RSK1 with different affinities, albeit following an ordered affinity ranking with conserved relative $K_D$ ratios. Remarkably, 14-3-3 isoforms obey the same hierarchy when binding to most of their established targets, as supported by literature and a recent human complexome map. This knowledge allows predicting proportions of 14-3-3 isoforms engaged with phosphoproteins in various tissues. Notwithstanding their individual functions, cellular concentrations of 14-3-3 may be collectively adjusted to buffer the strongest phosphorylation outbursts, explaining their expression variations in different tissues and tumors.

[1] Equipe Labellisee Ligue 2015, Department of Integrated Structural Biology, Institut de Genetique et de Biologie Moleculaire et Cellulaire (IGBMC), INSERM U1258/CNRS UMR 7104/Universite de Strasbourg, Illkirch, France. [2] A.N. Bach Institute of Biochemistry, Federal Research Center of Biotechnology of the Russian Academy of Sciences, Moscow, Russia. [3] These authors contributed equally: Gergo Gogl, Kristina V. Tugaeva. ✉email: goglg@igbmc.fr; traveg@igbmc.fr; nikolai.sluchanko@mail.ru

4-3-3 proteins recognize protein partners phosphorylated at serine or threonine in certain sequence motifs in all eukaryotic organisms. The seven human 14-3-3 isoforms, individually named β, γ, ε, ζ, η, σ, and τ (beta, gamma, epsilon, zeta, eta, sigma, and tau)[1], are distinct gene encoded paralogs that are highly similar in sequence and in their phosphopeptide-recognition mode, yet display different expression patterns across tissues[2,3]. 14-3-3 proteins are highly abundant in most human tissues, where several 14-3-3 isoforms are systematically found among the top 1% of the ~20,000 human gene-encoded proteins[3]. For instance, according to the Protein Abundance Database, PAXdb[3], the cumulated seven 14-3-3 isoforms are within the five most abundant protein species in platelets.

14-3-3 proteins have a highly conserved dimeric all-helical structure[1,4]. Each monomer is formed by a bundle of nine anti-parallel helices: the N-terminal α1-α4 helices comprise a dimerization zone and a bottom of the cup-shaped dimer, whose walls are built by the C-terminal α-helices[5]. Each monomer features a well-conserved amphipathic groove, a much less conserved convex solvent-exposed face, and a disordered C-terminal tail[1,4]. 14-3-3 proteins can form homodimers or heterodimers comprising two different isoforms[4,6]. According to various observations in vitro and in cells, 14-3-3σ preferentially homodimerizes, 14-3-3ε preferentially heterodimerizes (with any isoform except 14-3-3σ), whereas other isoforms tend to indifferently homodimerize or heterodimerize[4]. Heterodimerization preferences can be explained, at least in part, by the number of intermolecular salt bridges that can occur at the dimer interface[4]. However, a structure of a 14-3-3 heterodimer is still awaited, and so is a comprehensive study of homo- and heterodimerization affinity and/or kinetic constants of all isoforms. The cellular proportions of homo- and heterodimers are likely to vary depending on numerous factors such as the cellular concentrations of each isoform, their turnover rates, localization and post-translational modifications, which in turn will all vary depending on cell type and cellular status.

14-3-3 isoforms all have the ability to bind phosphopeptides[7,8]. Each monomer can bind one phosphopeptide via its amphipathic groove. Consequently, a 14-3-3 dimer can bind two phosphosites simultaneously. Those can originate from two different regions of the same protein, or from two different proteins. Phosphorylated 14-3-3-binding sequences usually correspond to internal motifs I RSX(pS/pT)X(P/G) or II RXY/FX(pS/pT)X(P/G)[8], or to the C-terminal motif III (pS/pT)X$_{0-2}$-COOH[9,10], where pS/pT denotes phosphorylated serine or threonine and X denotes any amino acid. The regulation by 14-3-3 binding typically protects 14-3-3 targets from dephosphorylation, thereby affecting their activities, their interactions with other proteins, their turnover, and intracellular localization[11]. 14-3-3 proteins are indispensable in a diversity of processes such as apoptosis, cell cycle, or signal transduction[1,12]. They are involved in neurodegenerative disorders, viral infection, and cancer, often representing promising drug targets[13].

14-3-3 isoforms also directly interact with several viral proteins[14], such as the E6 oncoprotein of high-risk mucosal human papillomaviruses (hrm-HPV)[15–17] responsible for genital cancers (cervix, anus) and a growing number of head-and-neck cancers[18,19]. E6 is one of the two main early-expressed HPV oncoproteins. In HPV-transformed cells, E6 interacts with numerous host proteins[20] to counteract apoptosis, alter differentiation pathways, polarity and adhesion properties, and thereby sustain cell proliferation[21,22]. Inhibition of E6 in HPV-positive cell lines results in cell growth arrest and induces apoptosis or rapid senescence[23–26]. All hrm-HPV E6 proteins harbor a phosphorylatable dual-specificity C-terminal motif[27] (Fig. 1a). In its unphosphorylated state, this is a PDZ domain-binding motif

(PBM) that mediates E6 binding to a range of cognate host proteins regulating cell polarity, adhesion, differentiation, or survival[17]. When the motif is phosphorylated, E6 proteins, in particular those of hrm-HPV 16, 18, and 31, acquire the capacity to bind 14-3-3[15,16,28].

Here we study the structural basis and druggability of 14-3-3 binding to E6 oncoproteins of four tumorigenic HPV types by a combination of crystallography, binding assays, and mutagenesis. We show that the seven 14-3-3 isoforms bound phospho-PBMs of E6 proteins and of the unrelated human RSK kinase with different affinities, albeit obeying a hierarchized profile with conserved relative $K_D$ ratios. This hierarchy turns out to be a general feature of the interaction of 14-3-3 isoforms with most of their targets, supported by literature and a recently released proteome-wide human complexome map[29]. Using this knowledge, we built a predictor that estimates the proportions of 14-3-3 isoforms engaged with phosphoproteins in various human tissues, cell lines, or tumors.

## Results

**E6 PBMs show parallel binding profiles to 14-3-3 isoforms.** Among all 225 HPV E6 proteins curated in the PaVE database (https://pave.niaid.nih.gov/, Accessed December 9, 2020), 31 E6 proteins from mucosal α-genera HPV possess a C-terminal PBM with the class 1 consensus (X(S/T)X(L/V/I/C)-COOH, where X is any amino acid residue[30,31]. E6 PBMs are phosphorylatable by protein kinases at their conserved antepenultimate S/T residue[15,16,32]. This phosphosite is preceded by arginine residues in most of the HPV E6 PBM sequences with recognizable basophilic kinase substrate consensus motifs, R(X/R)X(S/T) and RXRXX(S/T)[33,34]. The E6 PBMs can be classified in three subgroups: subgroups 1 and 2 prone to phosphorylation by the basophilic kinases and orphan subgroup 3 with a less predictable phosphorylation propensity (Supplementary Fig. 1). In line with earlier observations[15,28,35], the phospho-PBM sequences from subgroups 1 and 2 ideally match the C-terminal 14-3-3-binding motif III[9] (Fig. 1a and Supplementary Table 1).

The seven full-length human 14-3-3 isoforms, produced as fusions to a maltose-binding protein (MBP), were analyzed for their interaction with four phospho-PBMs from E6 proteins of HPV types 16, 18, 33, and 35 belonging to subgroups 1 and 2 (as defined in Supplementary Fig. 1). For comparison, we also measured two non-viral phospho-PBMs originating from protein kinase RSK1[28]. We used a competitive fluorescence polarization (FP) assay that measures the displacement of a fluorescent tracer phosphopeptide (here, derived from the HSPB6 protein) bound beforehand to 14-3-3, by an increasing amount of the peptide of interest. All binding curves are shown in Supplementary Fig. 2a.

All phospho-PBMs (p16E6, p18E6, p33E6, p35E6, RSK1_-1P, and RSK1_-2P) detectably bound to 14-3-3 proteins, in sharp contrast to their unphosphorylated counterparts. The interactions between E6 phospho-PBMs and 14-3-3 proteins spanned very wide affinity ranges, from just below 1 μM (p33E6–14-3-3γ) to above 300 μM (Fig. 1b and Supplementary Fig. 2a). Such large binding affinity differences are noteworthy since the four E6 PBM sequences are very similar (Fig. 1a), and all 14-3-3 isoforms share highly conserved phosphopeptide-binding grooves[1,4].

Remarkably, the six phospho-PBMs obeyed a consistent hierarchized profile in their relative binding preferences toward the seven 14-3-3 isoforms, albeit with an overall shift in affinity from one peptide to another. For each phosphopeptide, the seven 14-3-3 isoforms systematically clustered as four groups of decreasing affinity, in a conserved order from the strongest to the weakest phospho-PBM binder: gamma, eta, zeta/tau/beta, and epsilon/sigma (γ, η, ζ/τ/β, and ε/σ) (Fig. 1b). These conserved

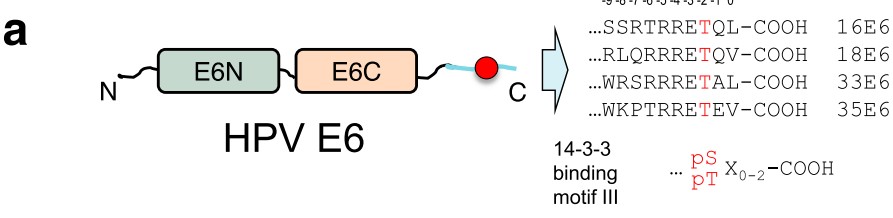

**a**

HPV E6

```
                -9 -8 -7 -6 -5 -4 -3 -2 -1 0
...SSRTRRETQL-COOH  16E6
...RLQRRRETQV-COOH  18E6
...WRSRRRETAL-COOH  33E6
...WKPTRRETEV-COOH  35E6
```

14-3-3 binding motif III

$...\begin{matrix} pS \\ pT \end{matrix} X_{0-2}$–COOH

**b**

| | $K_D$ (µM) ± std | | | | | | |
|---|---|---|---|---|---|---|---|
| | **14-3-3γ** | **14-3-3η** | **14-3-3ζ** | **14-3-3τ** | **14-3-3β** | **14-3-3ε** | **14-3-3σ** |
| **pHPV33E6** | 0.9 ± 0.3 | 2.0 ± 0.2 | 2.7 ± 0.4 | 3.5 ± 0.3 | 4.1 ± 0.7 | 10.3 ± 2.4 | 6.8 ± 1.1 |
| **pHPV18E6** | 11.1 ± 0.6 | 23.6 ± 6.2 | 22.4 ± 1.3 | 37.8 ± 6.0 | 42.2 ± 6.3 | 101 ± 20 | 139 ± 24 |
| **pHPV16E6** | 37.2 ± 3.0 | 80.5 ± 16.9 | 73.6 ± 19.1 | 144 ± 28 | 159 ± 45 | >300 | >300 |
| **pHPV35E6** | 125 ± 6 | 163 ± 19 | 191 ± 18 | 233 ± 14 | >300 | >300 | >300 |
| **pHPV35E6**$_{T-6R}$ | 71.2 ± 6.7 | 85.0 ± 20.6 | 139 ± 21 | 194 ± 37 | 179 ± 29 | 270 ± 20 | >300 |
| **pHPV35E6**$_{E-1A}$ | 9.3 ± 0.7 | 12.4 ± 1.5 | 17.6 ± 3.6 | 21.5 ± 1.2 | 20.2 ± 1.4 | 31.9 ± 4.2 | 44.1 ± 13.3 |
| **pHPV35E6**$_{T-6R, E-1A}$ | 3.3 ± 0.2 | 4.8 ± 0.5 | 7.3 ± 0.4 | 5.8 ± 0.4 | 7.1 ± 0.5 | 13.0 ± 2.0 | 18.9 ± 1.9 |
| **RSK1_-1P** | 0.3 ± 0.0 | 0.5 ± 0.1 | 1.0 ± 0.1 | 0.8 ± 0.1 | 0.7 ± 0.0 | 1.6 ± 0.2 | 1.7 ± 0.2 |
| **RSK1_-2P** | 0.2 ± 0.0 | 0.7 ± 0.1 | 1.5 ± 0.1 | 0.5 ± 0.1 | 1.0 ± 0.0 | 1.2 ± 0.2 | 3.6 ± 0.3 |

| | **14-3-3γ** | **14-3-3η** | **14-3-3ζ** | **14-3-3τ** | **14-3-3β** | **14-3-3ε** | **14-3-3σ** |
|---|---|---|---|---|---|---|---|
| **pHPV33E6** | | | | | | | |
| **pHPV18E6** | | | | | | | |
| **pHPV16E6** | | | | | | | |
| **pHPV35E6** | | | | | | | |

**c**

$\Delta\Delta G_{av}$ = -0.71±1.19  -1.50±0.63  -0.34±0.66  -0.14±1.39  -0.87±0.78  -1.35±0.82 (kJ/mol)

14-3-3σ < 14-3-3ε < 14-3-3β < 14-3-3τ < 14-3-3ζ < 14-3-3η < 14-3-3γ

$\Delta\Delta G_{av}$ = -2.06±0.67  -3.09±0.15  -5.98±0.65 (kJ/mol)

HPV35E6 < HPV16E6 < HPV18E6 < HPV33E6

**Fig. 1 E6 PBMs reveal parallel binding profiles to human 14-3-3 isoforms. a** Exemplary phosphorylatable C-terminal E6 PBMs from high-risk mucosal HPV types contain the 14-3-3-binding motif III[9]. The domains of the E6 protein are shown by green (E6N) and beige (E6C) colors, the C-terminal tail containing the phosphorylatable residue (red circle) is cyan. The positions are numbered above the sequences, according to conventional PBM numbering, with the phosphorylatable antepenultimate residue (position −2) indicated by red. **b** Affinities of four selected HPV E6 phospho-PBMs, p35E6 mutant variants, and RSK1 phosphopeptides toward the seven human 14-3-3 isoforms as determined by fluorescence polarization using FITC-labeled HSPB6 phosphopeptide as a tracer. Apparent $K_D$ values determined from competitive FP experiments are presented. The heatmap representation of the data on **b** shows the affinity trends in the interaction profiles between 14-3-3 isoforms and four HPV E6 phospho-PBMs from strongest (red) to weakest (white). Protein names are boldfaced for clarity. **c** Averaged $\Delta\Delta G$ values between 14-3-3 isoforms and E6 phospho-PBM pairs, calculated based on their observed order of binding affinities (from weakest to strongest). Individual $K_D$ values from Supplementary Fig. 2 were first converted into $\Delta G$ values (at $T = 295$ K; excluding cases when $K_D > 300$ µM), then average $\Delta\Delta G$ values ($\Delta\Delta G_{av}$) were calculated between the indicated motifs/isoforms. Standard deviation (std) values are indicated at each number on **b** and **c** (each time, three independent measurements). All binding data are provided as Supplementary Data File 1.

relative affinity shifts can be quantified by calculating, for two distinct 14-3-3 isoforms, their differences of free energy of binding ($\Delta\Delta G$) toward each individual phosphopeptide, then calculating the average difference ($\Delta\Delta G_{av}$) with its standard deviation (Fig. 1c). Between the strongest and the weakest binders (isoforms γ and σ, respectively) the average phosphopeptide-binding energy difference is $\Delta\Delta G_{av} = -5.1 \pm 1.3$ kJ/mol, roughly corresponding to a 11-fold $K_D$ ratio.

The seven 14-3-3 isoforms also showed consistent profiles in their relative binding preferences toward the four E6 phospho-PBMs. For each 14-3-3 isoform, the four phospho-PBMs systematically rank the same way from the strongest to the weakest binder: p33E6, p18E6, p16E6, and p35E6 (Fig. 1c). The average 14-3-3 binding free energy difference between p33E6 and p35E6 was $\Delta\Delta G_{av} = -10.9 \pm 0.7$ kJ/mol, roughly corresponding to a 100-fold $K_D$ ratio.

Of note, the presence of the MBP tag did not affect the relative affinity differences observed for the 14-3-3 isoforms since selected

untagged 14-3-3 isoforms obeyed the same trend from the strongest to the weakest (tau/beta and epsilon) (Supplementary Fig. 2c) observed for the MBP-tagged variants (Fig. 1b, c). The untagged 14-3-3 variants also preserved toward the four selected HPV E6 PBMs the preferences observed in more detail for the full-length MBP-tagged 14-3-3 isoforms (Fig. 1).

**Atomic structure reveals the 14-3-3ζ–18E6 PBM interface.** To get structural insight into the 14-3-3ζ interaction with 18E6 PBM, we determined a crystal structure of the 14-3-3ζ–18E6 phospho-PBM complex at a 1.9 Å resolution using a previously reported chimeric fusion strategy[36] (Table 1, Fig. 2, and Supplementary Figs. 3 and 4). The phosphopeptide establishes multiple polar interactions with the basic pocket in the amphipathic groove of 14-3-3 (Supplementary Fig. 5), largely reminiscent of previously solved structures of 14-3-3–phosphopeptide complexes[8,36]. The conformation of 18E6 phosphopeptide bound to 14-3-3ζ within

**Table 1 Crystallographic statistics.**

| | 14-3-3ζ-18E6 chimera | 14-3-3ζ-18E6 chimera + FSC |
|---|---|---|
| **Data collection** | | |
| Wavelength | 1.00 | 1.00 |
| Resolution range | 39.26–1.90 (1.95–1.90) | 38.05–1.85 (1.90–1.85) |
| Space group | P 21 21 21 | P 21 21 21 |
| Unit cell (a, b, c, α, β, γ) | 72.4, 78.5, 90.3, 90, 90, 90 | 73.2, 76.1, 89.0, 90, 90, 90 |
| Total reflections | 547783 (37726) | 557315 (41375) |
| Unique reflections | 41285 (2986) | 41927 (3038) |
| Multiplicity | 13.3 (12.6) | 13.3 (13.6) |
| Completeness (%) | 100 (100) | 97.1 (96.5) |
| Mean $I/\sigma(I)$ | 13.2 (1.4) | 12.4 (1.4) |
| R-meas | 16.6 (205) | 16.3 (216) |
| $CC_{1/2}$ | 99.9 (54.7) | 99.8 (58.4) |
| **Refinement** | | |
| R-work | 0.18 | 0.19 |
| R-free | 0.21 | 0.22 |
| Number of non-hydrogen atoms | 4468 | 4589 |
| Macromolecules | 4034 | 4004 |
| Ligands | 24 | 102 |
| Solvent | 410 | 483 |
| Protein residues | 481 | 482 |
| RMS (bonds) | 0.006 | 0.006 |
| RMS (angles) | 0.83 | 0.71 |
| Ramachandran favored (%) | 99.36 | 98.51 |
| Ramachandran allowed (%) | 0.64 | 1.49 |
| Ramachandran outliers (%) | 0 | 0 |
| Rotamer outliers (%) | 1.65 | 2.13 |
| Clashscore | 6.68 | 3.17 |
| Average B-factor | 33.39 | 29.6 |
| Macromolecules | 32.42 | 28.49 |
| Ligands | 62.93 | 32.67 |
| Solvent | 41.20 | 38.11 |
| Number of TLS groups | 15 | 11 |
| PDB ID | 6ZFD | 6ZFG |

the chimera is practically identical (RMSD = 0.17 Å upon superimposition of Cα atoms of the peptides) to the 14-3-3σ-bound conformation of a synthetic 16E6 phosphopeptide reported very recently at a lower resolution (Fig. 2b)[28]. The observed conservation of most interface contacts within the two complexes suggests that these crystal structures can serve as templates to build accurate homology models of 14-3-3 complexes for other E6 phospho-PBMs or, more generally, other C-terminal motif III peptides phosphorylated on the antepenultimate position.

Nonetheless, a few noteworthy differences appear in a subset of the crystallographic conformers of 14-3-3/16E6 and 14-3-3/18E6 complexes. On the one hand, in one of the four conformers observed in the asymmetric unit of the 14-3-3σ/16E6 crystal, the side chains of Arg –7 (Gln in 18E6) and Glu –3 form an additional in-cis salt bridge (Fig. 2b). On the other hand, Arg –6 of 18E6 (Thr in 16E6) mediates a bipartite interaction with 14-3-3 in most of the observed conformers. It simultaneously interacts with the carbonyl of Asp223 and participates in a water-mediated interaction with Asn224 (Fig. 2b and Supplementary Fig. 5).

**Rational design rescues the weakest E6–14-3-3 interaction.** Next, we investigated possible causes of the remarkable 14-3-3

binding affinity differences observed between the four E6 phospho-PBMs.

In principle, the affinity of a series of variant peptides for a given protein may be modulated by two types of atomic contacts: intermolecular and intramolecular contacts within the formed complexes, and intramolecular contacts in the free unbound peptides.

As concerns contacts within the 14-3-3/E6 complexes, the crystal structures have shown that Arg –6 can mediate more interactions than Thr –6 with the generic 14-3-3 interface (Fig. 2b). Interestingly, position –6 is an Arg in the two strongest 14-3-3 binders (18E6 and 33E6) versus a Thr in the weakest ones (35E6 and 16E6).

As concerns possible contacts within the unbound peptides, we noticed that all E6 phospho-PBMs have a delicate charge distribution, with an acidic C-terminal segment (that includes the C-terminal -COOH group and the natural acidic or phosphorylated residues) and a basic N-terminal segment (that is also involved in recognition by kinases). These local charged segments may form, within the unbound phosphopeptide, transient in-cis interactions often referred to as charge clamps[37]. We speculated that Glu –1 in p35E6, the weakest 14-3-3 binder, might participate in such a charge clamp, thereby disfavoring its binding to 14-3-3.

To address these potential mechanisms, we synthesized three variants of the weakest 14-3-3 binder, p35E6. The first variant contained a T-6R substitution, which in principle could allow a more stable bound conformation, but may also stabilize charge clamps in the free form of the motif. The second variant contained an E-1A substitution, which in principle could destabilize in-cis charge clamps. A third variant contained both substitutions. All substitutions turned out to reinforce the binding affinities of 35E6 without altering the apparent preferences of the different 14-3-3 isoforms (Figs. 1b and 2c). Taken individually, T-6R moderately increased binding ($\Delta\Delta G_{av} = -1.1 \pm 0.5$ kJ/mol, 1.5-fold $K_D$ ratio), while E-1A strongly reinforced it ($\Delta\Delta G_{av} = -5.1 \pm 0.2$ kJ/mol, 11-fold $K_D$ ratio). When combined, the two substitutions synergistically increased binding ($\Delta\Delta G_{av} = -8.7 \pm 0.4$ kJ/mol, 35-fold $K_D$ ratio), thereby turning p35E6 into a strong 14-3-3 binder, just below p33E6 (Figs. 1b and 2c). These results indicate that the two above-stated mechanisms act in combination to generate the wide 14-3-3-binding affinity range displayed by distinct E6 phospho-PBMs despite their high sequence conservation.

**The 14-3-3/E6 PBM interaction is druggable by fusicoccin.** Fusicoccin (FSC) is a commonly used stabilizer of 14-3-3 complexes, when its insertion within the 14-3-3/phosphopeptide interface is allowed by phosphopeptide side chains of the amino acids in downstream positions relative to the phospho-residue[38–41]. This is especially the case with motif III phospho-peptide complexes of 14-3-3 having only one residue after the phosphosite[39,41,42]. Nevertheless, the effect of FSC on interaction of longer motif III phosphopeptides with 14-3-3 is less characterized (Supplementary Table 1).

We performed FP experiments to measure equilibrium binding affinity constants of complexes between the four HPV E6 phosphopeptides and 14-3-3 isoforms ζ and γ, in the presence of FSC (Fig. 3a and Supplementary Fig. 2b). The addition of FSC consistently decreased by 1.5- to 2-fold the affinities of all eight interactions ($\Delta\Delta G_{av} = -1.3 \pm 0.5$ and $-1.8 \pm 0.4$ kJ/mol for ζ and γ, respectively) without altering the apparent preferences of the different peptides (Fig. 3a, b).

Next, we used a soaking approach to crystallize the ternary 14-3-3ζ/18E6 PBM/FSC complex and solved its structure at 1.85 Å

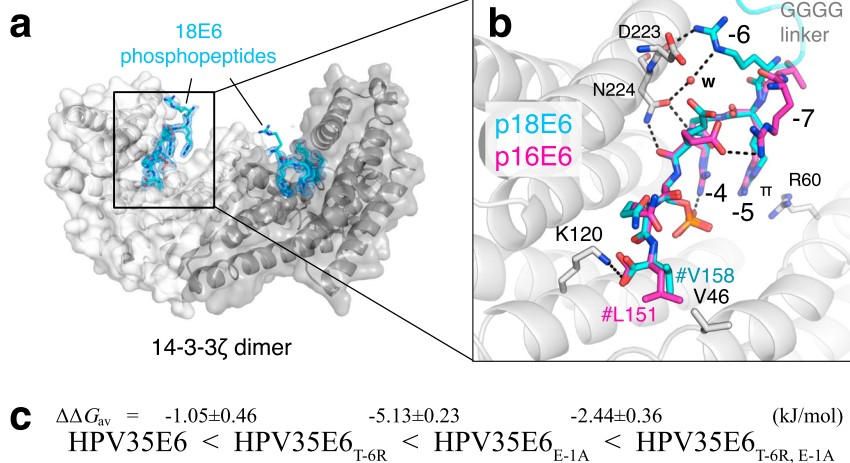

**c** $\Delta\Delta G_{av}$ = -1.05±0.46 -5.13±0.23 -2.44±0.36 (kJ/mol)

$$\text{HPV35E6} < \text{HPV35E6}_{\text{T-6R}} < \text{HPV35E6}_{\text{E-1A}} < \text{HPV35E6}_{\text{T-6R, E-1A}}$$

**Fig. 2 Structural basis for the 14-3-3ζ/phospho-18E6 PBM interaction. a** An overall view on the 14-3-3ζ dimer (subunits are in tints of gray) with two bound 18E6 phosphopeptides (cyan sticks, shown with the 2Fo-Fc electron density maps contoured at 1σ). **b** An overlay of the two 14-3-3 bound phosphopeptides from 16E6 (PDB ID: 6TWZ; magenta sticks) and 18E6 (this work; cyan sticks) showing the similarity of the conformation. # denotes the C-terminus (-COOH). w—water molecule, π—π-stacking interaction. Key positions are numbered according to the PBM convention. **c** Averaged $\Delta\Delta G$ values between 14-3-3 isoforms and 35E6 phospho-PBM pairs, calculated based on their observed order of binding affinities (from weakest to strongest). Individual $K_D$ values from Supplementary Fig. 2 were first converted into $\Delta G$ values (at $T = 295$ K; excluding cases when $K_D > 300$ μM), then average $\Delta\Delta G$ values ($\Delta\Delta G_{av}$) were calculated between the indicated HPV35E6 motifs.

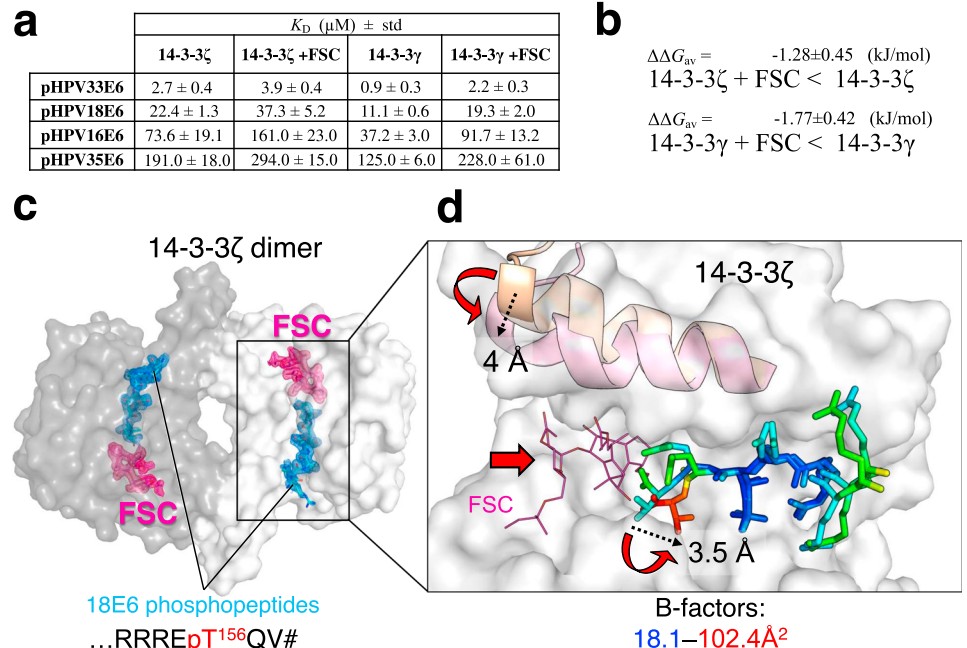

**a**

| | $K_D$ (μM) ± std | | | |
|---|---|---|---|---|
| | **14-3-3ζ** | **14-3-3ζ +FSC** | **14-3-3γ** | **14-3-3γ +FSC** |
| **pHPV33E6** | 2.7 ± 0.4 | 3.9 ± 0.4 | 0.9 ± 0.3 | 2.2 ± 0.3 |
| **pHPV18E6** | 22.4 ± 1.3 | 37.3 ± 5.2 | 11.1 ± 0.6 | 19.3 ± 2.0 |
| **pHPV16E6** | 73.6 ± 19.1 | 161.0 ± 23.0 | 37.2 ± 3.0 | 91.7 ± 13.2 |
| **pHPV35E6** | 191.0 ± 18.0 | 294.0 ± 15.0 | 125.0 ± 6.0 | 228.0 ± 61.0 |

**b**

$\Delta\Delta G_{av}$ = -1.28±0.45 (kJ/mol)

$$\text{14-3-3ζ} + \text{FSC} < \text{14-3-3ζ}$$

$\Delta\Delta G_{av}$ = -1.77±0.42 (kJ/mol)

$$\text{14-3-3γ} + \text{FSC} < \text{14-3-3γ}$$

**Fig. 3 The 14-3-3ζ/18E6 PBM interaction is druggable by FSC. a** Affinities of four selected HPV E6 phospho-PBMs toward human 14-3-3ζ and 14-3-3γ in the absence and presence of FSC as determined by FP using FITC-labeled HSPB6 phosphopeptide as a tracer. Protein names are boldfaced for clarity. Apparent $K_D$ values determined from competitive FP experiments are presented. Standard deviation (std) values are indicated. The binding curves are shown in Supplementary Fig. 2. **b** Averaged $\Delta\Delta G$ values between 14-3-3–E6 phospho-PBM pairs in the absence or presence of FSC, calculated based on their observed order of binding affinities (from weakest to strongest). Individual $K_D$ values from Supplementary Fig. 2 were first converted into $\Delta G$ values (at $T = 295$ K; excluding cases when $K_D > 300$ μM), then average $\Delta\Delta G$ values ($\Delta\Delta G_{av}$) were calculated for the E6-binding affinity changes of the indicated 14-3-3 isoforms in the absence or presence of FSC. **c** An overall view on the ternary complex between 14-3-3ζ (subunits are shown by surface using two tints of gray), 18E6 phosphopeptide (cyan sticks), and FSC (pink sticks). FSC was soaked into the 14-3-3ζ–18E6 chimera crystals. 2Fo-Fc electron density maps contoured at 1σ are shown for the phosphopeptide and FSC only. **d** The effect of FSC binding. Conformational changes in the 9th α-helix of 14-3-3 and in the C-terminal part of the 18E6 phosphopeptide upon FSC binding are shown by red arrows, a significant rise of the local B-factors of the phosphopeptide is shown using a gradient from blue to red as indicated. The amplitudes of the conformational changes are indicated in Å by dashed arrows.

resolution (Fig. 3c, Table 1, and Supplementary Figs. 4 and 5). While FSC binding in the well-defined cavity did not disrupt the overall assembly (Fig. 3c and Supplementary Fig. 4), it induced a hallmark ~4 Å closure of the last α-helix of 14-3-3ζ (Fig. 3d) as observed for other 14-3-3 complexes containing FSC[43]. Also, FSC binding reoriented the C-terminal carboxyl group and caused local destabilization of the very C-terminal residues of the phosphopeptide, increasing their temperature factors and dispersing the local electron density (Fig. 3d and Supplementary Fig. 5). As a result of FSC binding, the water network around the phospho-PBM C-terminus significantly changed (Supplementary Fig. 6).

Nevertheless, the simultaneous binding of FSC and E6 PBM in the amphipathic groove of 14-3-3 indicates that such ternary complex can be used as a starting point to design both stabilizers and inhibitors of 14-3-3/E6 interactions.

**Hierarchized peptide-affinity profiles are a general feature of human 14-3-3 isoforms.** Former studies have measured the binding of the seven human 14-3-3 isoforms to unrelated phospho-motifs derived from Cystic Fibrosis Transmembrane Conductance Regulator (CFTR), Leucine-Rich Repeat Kinase 2 (LRRK2), Potassium channel subfamily K members (TASK1/3), C-Raf, the p65 subunit of the NF-κB transcription factor, and from Ubiquitin carboxyl-terminal hydrolase 8 (USP8), representing a wide variety of different 14-3-3-binding motifs, including C-terminal, internal, monovalent, or divalent motifs[40,44–48]. These phosphorylated motifs from different origins have a strikingly wide affinity range, spanning from low nanomolar to low millimolar detectable dissociation constants (Fig. 4a–c). For instance, for 14-3-3γ, the $K_D$ ratio between the strongest and the weakest-binding phosphopeptide is almost 625-fold in the present work, and 39,000-fold when taking into account affinities from the literature (Fig. 4b).

Conversely, the hierarchized relative binding profile of the seven human 14-3-3 isoforms observed herein for E6 and RSK1 phosphopeptides is remarkably confirmed in most published data that have also measured affinities for all these seven isoforms[40,44–48], with 14-3-3γ and 14-3-3η consistently being the strongest binders and 14-3-3σ and 14-3-3ε being the weakest binders, independently of the nature of the target motif (Fig. 4c). Furthermore, the average maximal $K_D$ ratio between the strongest-binding and the weakest-binding 14-3-3 in the literature is around 12-fold, like in our present work (~11-fold).

Moving further, we wondered whether the observed trends would be conserved at a full proteome-wide scale. Recently, a massive parallel affinity-purification approach coupled to mass-spectrometry (AP-MS) was applied to decipher the complexomes of more than 10,000 recombinantly expressed bait proteins in two orthogonal cell lines[29]. We retrieved from the BioPlex 3 database (a compendium of the abovementioned complexome study) the numbers of detected interaction partners for each 14-3-3 isoform (Fig. 4d–g). In total, 547 unique proteins were detected as an interaction partner of at least a single 14-3-3 isoform. Out of those, 14-3-3γ and 14-3-3η had the highest number of interaction partners, followed by a second group including 14-3-3β, 14-3-3ζ, and 14-3-3τ, and a third group comprising 14-3-3σ and 14-3-3ε (Fig. 4d). Most of these interaction partners were found to bind more than a single 14-3-3 isoform (Fig. 4e, f). While the strongest-binding isoforms (γ and η) do not share ~30% of their interactome with the other isoforms, they interact with more than 85% of the binders of the mild-binding isoforms (β, ζ, and τ) and more than 90% of the binders of the weak-binding isoform 14-3-3ε. Indeed, out of the 75 detected binders of 14-3-3ε, only 1 (below 2% of the total) is unique to 14-3-3ε. By contrast, the other

weak-binding isoform, 14-3-3σ, has a distinct behavior. Out of its 51 detected binders, 26 interactions are unique to 14-3-3σ (above 50% of all its binders).

Furthermore, we observed a remarkable linear correlation ($R^2 = 0.91$) between the numbers of binders detected by the BioPlex project[29] for each 14-3-3 isoform, and their relative affinity ($\Delta\Delta G_{av}$) as compared to the strongest phosphopeptide-binding isoform, 14-3-3γ (Fig. 4g).

In the AP-MS experiments, interaction partners (and 14-3-3 proteins in particular) can be either baits or preys. Baits are recombinantly expressed in the cells using the same promoter, which should ensure a relatively even expression for all 14-3-3 isoforms. By contrast, the preys are proteins naturally expressed by the cells, so that the distinct 14-3-3 preys should be present in different amounts, depending on their intrinsic levels of expression in the host cells. In BioPlex, six out of seven 14-3-3 isoforms (with the exception of ε) were among the tested recombinant bait proteins. This allowed us to distinguish, among the 14-3-3 binders, those identified as baits retaining 14-3-3 preys, from those identified as preys retained by 14-3-3 baits. In both situations, the linear correlation of the numbers of 14-3-3 binders with the relative phosphopeptide-binding affinity scale of 14-3-3 proteins remained very strong ($R^2 = 0.96$ and $R^2 = 0.90$, respectively) (Supplementary Fig 7a).

In further support of these interrelations, the number of 14-3-3 prey-binding baits also indicated correlation with the affinity trend of 14-3-3 isoforms when using data from a recent independent study (https://sec-explorer.shinyapps.io/Kinome_interactions/ and[49]) that used AP-MS to uncover the interactions of more than 300 protein kinases ($R^2 = 0.64$) (Supplementary Fig. 7a).

The BioPlex database[29] also contains all peptide-spectrum matches (PSM) values for all the preys retained by each and every bait. PSM values bear information about the enrichment of prey proteins on resins carrying particular baits. The higher the PSM values measured for a given prey precipitated by a particular bait, the more enriched the prey, and, therefore, the stronger the affinity of the corresponding bait–prey complex. We retrieved and summed up the PSM values of the 114 preys captured in the BioPlex experiment[29] by five different 14-3-3 isoforms. These values show good agreement ($R^2 = 0.66$) with the affinity trends of the different 14-3-3 isoforms (Fig. 4h). Even more remarkably, when those 114 individual preys are ranked from their highest to lowest PSM values relative to 14-3-3γ (Supplementary Fig. 7b), one observes the same bi-directional decreasing intensity pattern as seen in our experiments (Fig. 1b) as well as in the low-throughput data from literature (Fig. 4a).

Altogether, these analyses indicate that both the numbers and the PSM enrichment values of partner proteins of 14-3-3 isoforms detected by proteome-wide interactomic studies are remarkably correlated with their relative phosphopeptide-binding affinity trends.

Notably, the level of the overall sequence divergence of 14-3-3 isoforms, using 14-3-3γ as a reference (γ < η < β ≈ ζ < τ < σ < ε; i.e., ε is the most divergent from γ), also correlates very well with their hierarchized affinity differences (Fig. 4i). However, the peptide-affinity trend cannot be explained merely by features of the phosphopeptide-binding regions of the 14-3-3 isoforms, which in fact are identical in all the seven human 14-3-3 proteins (Supplementary Fig. 8). Indeed, even the extreme isoforms on the peptide-affinity scale, 14-3-3γ and 14-3-3σ, have only minor sequence variations in their phosphopeptide-binding grooves and only at their periphery (Supplementary Fig. 8), which are unlikely to dictate the phosphopeptide-binding differences. Interestingly, the sequence divergence trend relative to 14-3-3γ (Fig. 4i) remains conserved when considering diverse sub-regions

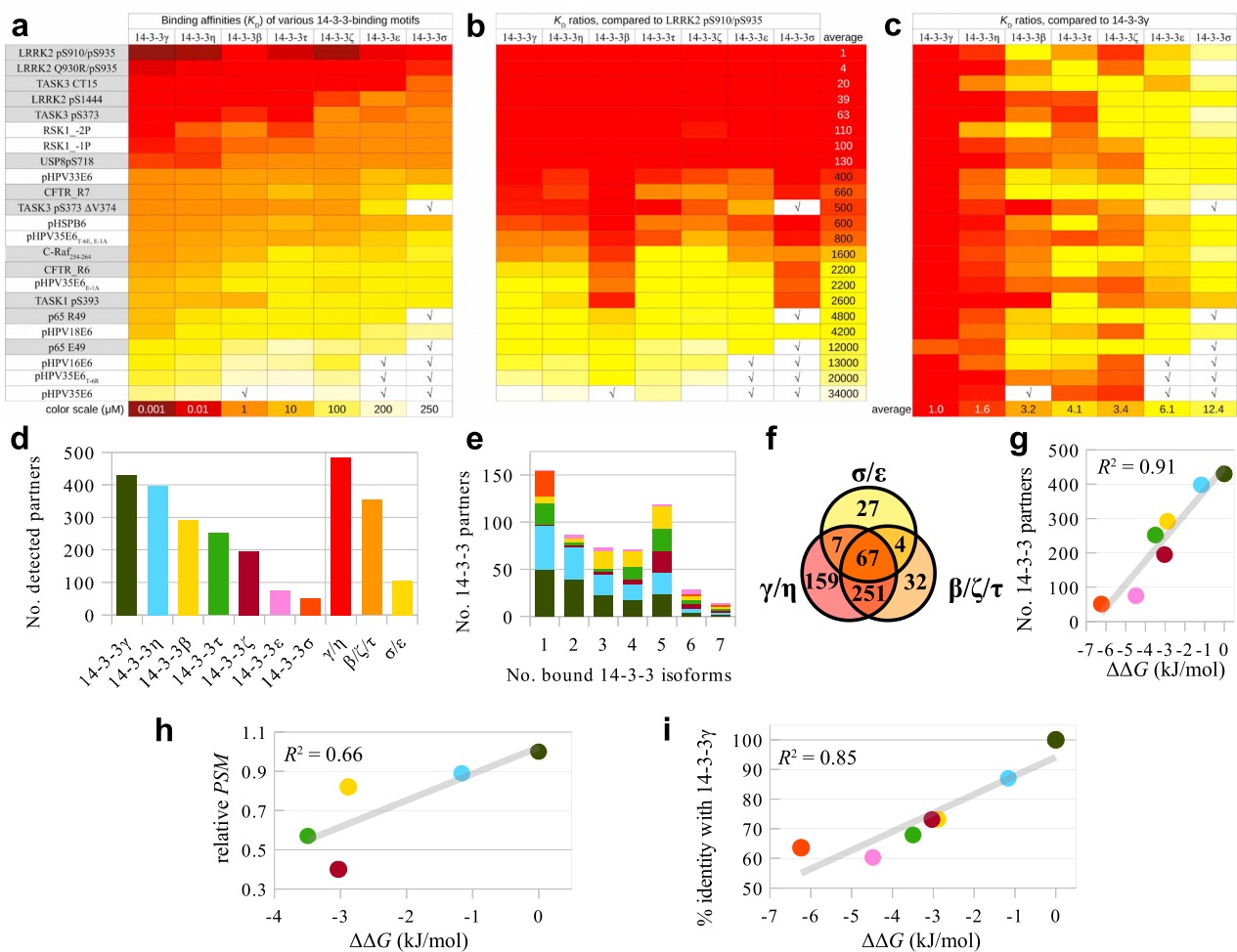

**Fig. 4 Hierarchized target binding by 14-3-3 isoforms is a general trend. a** Affinity maps of 14-3-3 interactions based on experimentally determined dissociation constants against the 14-3-3ome, as obtained in this study (white background) and in others[40,44–48] (gray background). The color scale is either based on affinity values or on $K_D$ ratios. √ denotes affinities weaker than the limit of quantitation of the experimental assays. **b** Same map as in **a**, normalized to the strongest 14-3-3-binding motif. An average 34,000-fold $K_D$ ratio is observed between the strongest and weakest 14-3-3-binding peptide. **c** Same map as in **a**, normalized to the strongest phosphopeptide-binding 14-3-3 isoform. Note that all peptides follow very similar affinity trends between the different 14-3-3 isoforms, with an average 12-fold $K_D$ ratio between the strongest and weakest-binding 14-3-3 isoform. **d** Number of unique partners detected according to the BioPlex database[29] for each 14-3-3 isoform, taken individually (left) or grouped in three subsets (right) following their relative affinity trends (strong, intermediate, and weak binders). **e** Number of 14-3-3 partners in BioPlex, which bound to 1, 2, 3, 4, 5, 6, or all 7 isoforms, respectively. Within each bar, the proportion of partners that bound to each individual isoform is indicated (same isoform color code as in **d**). **f** Venn diagram showing repartition of the 14-3-3 partners from BioPlex among the strong, medium, and weak phosphopeptide-binding subsets, defined as in **d**. **g** Correlation between the number of binders of 14-3-3 isoforms, according to BioPlex, and $\Delta\Delta G_{av}$ between the strongest phosphopeptide-binder, 14-3-3γ, and all individual isoforms (same color code as in **d**). $\Delta\Delta G$ values were calculated from the average $K_D$ ratios from **c**. **h** The average amounts of prey proteins from BioPlex (normalized to the amount captured using 14-3-3γ) that interact with at least five different 14-3-3 baits, deduced from their *PSM* values, also show a correlation with the $\Delta\Delta G_{av}$ values of the same proteins. **i** Correlation of the sequence identity of human 14-3-3 isoforms relative to 14-3-3γ with the $\Delta\Delta G_{av}$ values from **g**. Source data are provided as a Source Data file.

of the sequence (Supplementary Fig. 8). This indicates that the general target affinity differences arise from fine conformational effects spanning the entire structure, rather than a defined sub-region.

**Prediction of cellular 14-3-3/phosphotargets complexomes.** 14-3-3 proteins are highly expressed. Therefore, their abundances in all human tissues have been reliably quantified. According to the integrated whole human body dataset of the PAXdb (https://pax-db.org and[3]), 14-3-3ε is the 48th most abundant human protein (2479 ppm) and 14-3-3ζ is the 72nd (1680 ppm) out of 19,949 proteins. Considered as a whole, the cumulated seven 14-3-3 isoforms even rank within the top 20 (i.e., top 0.1%) most abundant human proteins. However, 14-3-3 isoforms are

not uniformly distributed across tissues. Each human cell type displays a specific distribution of the 14-3-3 family (Fig. 5a).

We took advantage of the quantified hierarchized affinity profile of 14-3-3 isoforms to build a predictor tool that estimates the fraction of a given phosphoprotein that is engaged with each distinct 14-3-3 isoform (Supplementary Data File 2). As an input, the predictor requires (i) the $K_D$ of that phosphoprotein for at least one 14-3-3 isoform, and (ii) the cellular concentrations of the seven 14-3-3 isoforms and of the phosphoprotein of interest. The concentrations of a given protein species in a given cell type can be roughly estimated from protein abundance databases (such as PAXdb[3]), by using a simple conversion rule (see Methods).

We used this approach to predict the proportions of each 14-3-3 isoform among the overall 14-3-3/phosphoprotein complexes

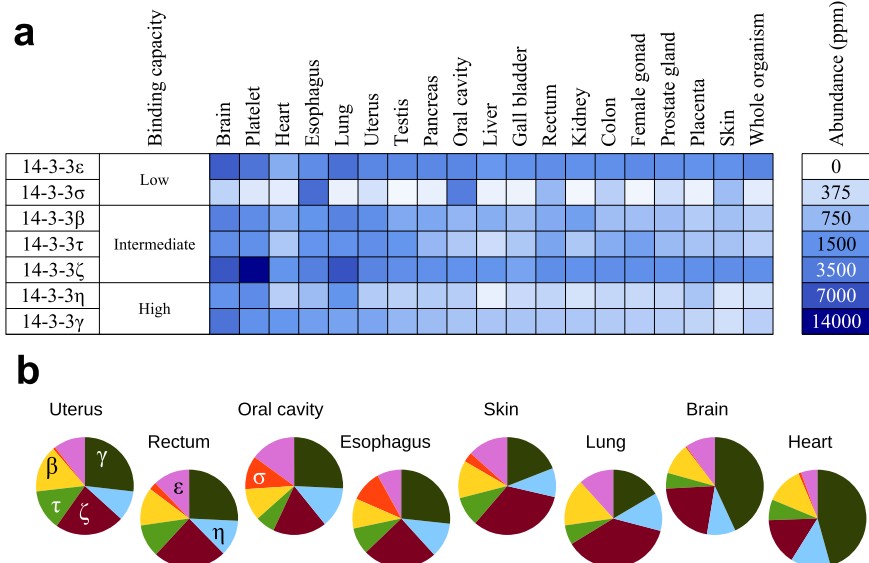

**Fig. 5 Cellular 14-3-3/phosphotarget complexomes. a** Abundance of the seven 14-3-3 isoforms across different human tissues and in the whole human organism, according to the PAXdb database (https://pax-db.org and [3]). Colors correspond to the protein abundances, according to the scale provided on the right. **b** Predicted proportions of 14-3-3-bound phosphoproteins that would be engaged with each individual isoform in different tissues, assuming that the majority of 14-3-3 molecules are available for interaction (same color code as in Fig. 4d). Source data are provided as a Source Data file.

formed in various tissues, including uterus, rectum, and oral cavity, which are all susceptible to hrm-HPV infection, as well as in five other organs (esophagus, skin, lung, brain, and heart) (Fig. 5b and Supplementary Fig. 9).

Conversely, the free fraction of each phosphoprotein depends on absolute affinity constants. HPV-positive cell lines have been estimated to produce an average of ~1 ng of E6 per $10^6$ cells, corresponding to an approximate intracellular concentration of 25 nM[50]. In a situation where the E6 PBM would be fully phosphorylated, we can estimate, using the predictor, that 87%, 35%, 14%, or 5% of phosphorylated 33E6, 18E6, 16E6 and 35E6, respectively, would be engaged in 14-3-3 complexes in cells containing the average 14-3-3 concentrations found in human cells (estimated from integrated human data in PAXdb database[3]) (Supplementary Fig. 9).

**Discussion**

E6 oncoproteins of all hrm-HPV types contain a conserved C-terminal PDZ-Binding Motif that can become a potential 14-3-3-binding motif upon phosphorylation[15,16,28] (Fig. 1a and Supplementary Fig. 1). Here, we initially set out to analyze the mechanistic and structural basis for the 14-3-3ζ binding to the 18E6 oncoprotein. Comparison to the previously solved complex between 14-3-3σ and HPV16 E6[28] revealed conserved binding principles (Fig. 2b) that are likely to be valid for most hrm-HPV E6/14-3-3 complexes. We also showed that the FSC molecule, a well-known modulator of 14-3-3 interactions, moderately destabilizes E6 binding to 14-3-3 (Fig. 3). This indicated that the hrm-HPV E6/14-3-3 complexes are in principle druggable.

The phosphorylated PBMs of four selected hrm-HPV E6 all detectably bound to 14-3-3 proteins, albeit with surprisingly wide affinity variations spanning a 100-fold $K_D$ range for different E6 PBMs binding to a given 14-3-3 isoform (Fig. 1). In the literature, interactions of phosphorylated peptides with 14-3-3 even cover a wider ~40,000-fold affinity range, from low nanomolar to low millimolar (Fig. 4). As shown in the present work, very modest sequence variations of a phosphopeptide can be sufficient to alter its unbound and/or bound states in a way that greatly impacts

binding affinity (Fig. 2c). Similar principles may govern 14-3-3-binding affinity variations of many other phosphopeptides.

Conversely, the seven 14-3-3 isoforms bound each E6 phosphopeptide following a conserved hierarchized profile, with an approximate 11-fold $K_D$ ratio between the strongest-binding and the weakest-binding 14-3-3 isoform. Remarkably, 14-3-3 proteins obey the same hierarchy when binding to most of their targets, as supported by our own data on RSK1 and HSPB6 peptides, by our literature curation[40,44–48], and by the unbiased proteome-wide complexome datasets such as the BioPlex 3 database (https://bioplex.hms.harvard.edu and[29]) or the human kinome interactome (https://sec-explorer.shinyapps.io/Kinome_interactions/ and[49]). Only 14-3-3σ may stand out as a partial exception to this rule. While displaying a low affinity to most 14-3-3 targets, it nonetheless binds to a small subset of proprietary targets that are not shared with other 14-3-3 isoforms (Fig. 4). This outlier character of 14-3-3σ has already been noticed in previous works dedicated to the structural and functional peculiarities of that isoform[51,52].

We took advantage of the hierarchized target-binding profiles of 14-3-3 isoforms to develop a prediction approach of the 14-3-3 complexome. This approach can compute, for a given cell population, the free and 14-3-3-bound fractions of any phosphoprotein whose cellular concentration and affinity for at least one 14-3-3 isoform are available. The concentration of host proteins can be inferred from the protein abundance databases such as PAXdb (https://pax-db.org and[3]), whereas the affinity to a 14-3-3 isoform can easily be obtained using state-of-the-art quantitative in vitro binding assays. While 14-3-3 proteins predominantly exist as dimers, the predictor deals with concentrations of 14-3-3 isoform monomers. The calculation assumes that the affinity of each monomer molecule toward a single phospholigand corresponds to the affinity measured for homodimers and is not influenced by the nature of the neighbor monomer, be it the same isoform (homodimeric species) or of another isoform (heterodimeric species). This assumption is plausible, considering the very high conservation of the amphipathic grooves of 14-3-3 proteins (Supplementary Fig. 8), responsible for ligand binding and facing each other in the dimeric structures. Anyhow, our

predictor should be mainly intended as a rough trend estimator to stimulate thinking and explore hypotheses, rather than an accurate descriptor of the actual precise proportions of 14-3-3 complexes in cells.

When applied to the rather weakly expressed HPV E6 proteins, predictions indicated that, in a cellular situation favoring E6 phosphorylation, phospho-E6 molecules should get fully engaged with the 14-3-3 pool for the strongest 14-3-3-binding E6 variants, and only partly engaged for the weaker ones. Such differences are likely to influence the dephosphorylation kinetics of phospho-E6 molecules from different HPV types, and the subsequent dynamics of cellular mechanisms involving PDZ-containing proteins targeted by E6. We also found that, in tissues susceptible to HPV infections, phosphorylated E6 would be complexed to distinct proportions of 14-3-3 proteins. In particular, phosphorylated E6 might be engaged with a higher proportion of 14-3-3σ in oral cavity, where this otherwise weakly expressed isoform is particularly abundant (Fig. 5).

14-3-3 proteins are abundant in all tissues, yet in variable amounts. It is also known that most tumors adjust their 14-3-3 concentrations, by altering the expression of at least one 14-3-3 isoform[53–55]. In all cell types, peaks of bulk phosphorylation occur, for instance at specific cell cycle steps or in reaction to changes in the extracellular environment[56,57]. It is tempting to speculate that, as previously proposed by others[58], 14-3-3 proteins, notwithstanding their individual functional specificities, may collectively provide a buffering system for intracellular signaling. In such a view, the cumulated concentrations of 14-3-3 are adjusted in each cell type for coping with the most acute phosphorylation outbursts possible in that very cell type. We notice that the highest concentrations of 14-3-3 in human cells are found in platelets (Fig. 5a). Indeed, platelet activation is a phenomenon known to involve powerful phosphorylation events[59].

To conclude, the present work opens novel avenues for interpreting, predicting and addressing in a quantitative and global manner the way that distinct 14-3-3 isoforms bind to pools of phosphorylated proteins and thereby modulate their activities.

## Methods

**Cloning, protein purification, and peptide synthesis**. Previously described chimeras contained the C-terminally truncated human 14-3-3σ (Uniprot ID P31947; residues 1-231, 14-3-3σΔC) bearing on its N terminus a His$_6$-tag cleavable by 3C protease and phosphorylatable peptides tethered to the 14-3-3σ C terminus by a GSGS linker[36]. The novel chimera was designed taking into account the following modifications. First, it contained the C-terminally truncated human 14-3-3ζ sequence (Uniprot ID P63104; residues 1-229, 14-3-3ζΔC) connected to the PKA-phosphorylatable 18E6 heptapeptide around Thr156. Second, the 14-3-3ζ core was modified to block Ser58 phosphorylation (S58A)[36,60]. Third, to improve crystallizability, the 14-3-3ζ sequence was mutated by introducing the $^{73}$EKK$^{75}$ → AAA and $^{157}$KKE$^{159}$ → AAA amino acid replacements in the highest-scoring clusters 1 and 2 predicted by the surface entropy reduction approach[61]. Finally, the linker was changed to GGGG to exclude its unspecific phosphorylation (Supplementary Fig. 3a).

cDNA of the 14-3-3ζ-18E6 chimera was codon-optimized for expression in *Escherichia coli* and synthesized by IDT Technologies (Coralville, Iowa, USA). The 14-3-3ζΔC gene was flanked by NdeI and AgeI restriction endonuclease sites to enable alteration of the 14-3-3 or E6 PBM peptide sequences. The entire 14-3-3ζ-GGGG-18E6 PBM construct was inserted into a pET28-his-3C vector[60] using NdeI and XhoI restriction endonuclease sites. The resulting vector was amplified in DH5α cells and verified using DNA sequencing in Evrogen (Moscow, Russia, www.evrogen.ru).

The assembled vector (Kanamycin resistance) was transformed into chemically competent *E. coli* BL21(DE3) cells for expression either in the absence or in the presence of the His$_6$-tagged catalytically active subunit of mouse PKA[60]. Protein expression was induced by the addition of isopropyl-β-thiogalactoside (IPTG) to a final concentration of 0.5 mM and continued for 16 h at 25 °C. The overexpressed protein was purified using subtractive immobilized metal-affinity chromatography (IMAC) and gel-filtration[36] (Supplementary Fig. 3b, c). The purified phosphorylated 14-3-3ζ-18E6 chimera revealed the characteristic downward shift on native PAGE compared to the unphosphorylated counterpart (Supplementary

Fig. 3d). Given the absence of PKA phosphorylation sites in the modified 14-3-3ζ core and the linker, this strongly indicated 18E6 phosphorylation by co-expressed PKA. The chimera was fully soluble and stable at concentrations above 20 mg/ml required for crystallization. Protein concentration was determined at 280 nm on a Nanophotometer NP80 (Implen, Germany) using extinction coefficient equal to 0.93 (mg/ml)$^{-1}$ cm$^{-1}$.

For affinity measurements, full-length human 14-3-3 constructs with an N-terminal MBP fusion were used. Plasmids containing the cDNAs of the full-length 14-3-3 isoforms ε, γ, and ζ were received from Prof. Lawrence Banks. cDNAs encoding other full-length 14-3-3 isoforms β, τ, η, and σ were obtained as codon-optimized for *E. coli* expression synthetic genes from IDT Technologies (Coralville, Iowa, USA). All 14-3-3 isoforms were fused via a three-alanine linker to the C terminus of a mutant MBP carrying the following amino acid substitutions facilitating crystallization: D83A, K84A, K240A, E360A, K363A, and D364A[62]. All resulting clones were verified by sequencing. The MBP-fused proteins were expressed in *E. coli* BL21(DE3) with IPTG induction. Proteins were affinity purified on an amylose column and were further purified by ion-exchange chromatography (HiTrap Q HP, GE Healthcare). Unfused 14-3-3 isoforms devoid of the flexible C-terminal tails (14-3-3βΔC, 14-3-3τΔC 14-3-3εΔC) were expressed using IPTG induction and purified by subtractive IMAC and SEC[40]. Protein concentrations were determined by UV spectroscopy. The samples were supplemented with glycerol and TCEP before aliquoting and freezing in liquid nitrogen.

HPV peptides (35E6: biotin-ttds-SKPTRRETEV; 16E6: biotin-ttds-SSRTRRETQL; 18E6: biotin-ttds-RLQRRRETQV; 33E6: biotin-ttds-SRSRRRETAL; p35E6: biotin-ttds-SKPTRREpTEV; p35E6 E-1A: biotin-ttds-SKPTRREpTAV; p35E6 T-6R: biotin-ttds-SKPRRREpTEV; p35E6 E-1A T-6R: biotin-ttds-SKPRRREpTAV; p16E6: biotin-ttds-SSRTRREpTQL; p18E6: biotin-ttds-RLQRRREpTQV; p33E6: biotin-ttds-SRSRRREpTAL) and RSK1 peptides (RSK1_-1P: biotin-ttds-RRVRKLPSTpTL and RSK1_-2P: biotin-ttds-RRVRKLPSpTTL) were chemically synthesized in-house on an ABI 443A synthesizer with Fmoc strategy. The fluorescently labeled HSPB6 (WLRRApSAPLPGLK) peptide (fpB6) was prepared by amino-terminal FITC labeling of the chemically synthesized peptide[28].

**Fluorescence polarization (FP) assay**. Unless otherwise stated, all phosphopeptide-binding measurements were performed using full-length 14-3-3 proteins fused to the C terminus of MBP, favoring optimal solubility and a standardized parallel purification scheme. Preliminary measurements using three unfused 14-3-3 constructs devoid of the flexible C-terminal tail (14-3-3βΔC, 14-3-3τΔC, 14-3-3εΔC), often absent from 14-3-3 in structural studies[40], showed comparable affinity values (Supplementary Fig. 2c), indicating that MBP had no influence on the relative affinity trends analyzed in this work.

FP was measured with a PHERAstar (BMG Labtech, Offenburg, Germany) microplate reader by using 485 ± 20 and 528 ± 20 nm band-pass filters (for excitation and emission, respectively). In direct FP measurements, a dilution series of the 14-3-3 protein was prepared in 96-well plates (96-well skirted PCR plate, 4ti-0740, 4titude, Wotton, UK) in a 20 mM HEPES pH 7.5 buffer containing 150 mM NaCl, 0.5 mM TCEP, 0.01% Tween 20, 50 nM fluorescently labeled fpB6 peptide, and 100 μM FSC, if indicated. The volume of the dilution series was 40 μl, which was later divided into three technical replicates of 10 μl upon transferring to 384-well micro-plates (low binding microplate, 384 well, E18063G5, Greiner Bio-One, Kremsmünster, Austria). In total, polarization of the probe was measured at eight different protein concentrations (whereas one contained no protein and corresponded to the free peptide). In competitive FP measurements, the same buffer was supplemented with the protein to achieve a complex formation of 60–80%, based on the direct titration. Then, this mixture was used for creating a dilution series of the unlabeled competitor (i.e., the studied peptides) and the measurement was carried out identically as in the direct experiment.

All FP experiments were done in triplicates. Analysis of FP experiments were carried out using ProFit, an in-house developed, Python-based fitting program[63]. ProFit utilizes a Monte Carlo approach to take into account experimental variability. It generates hundreds of simulated datasets, based on the experimental data variance and fits direct and competitive measurements in pairs. The experimental polarization window is first determined in the direct experiment, then this is either used as a fixed restraint in competitive fits or as a reference value to validate the result of unrestrained fits. In cases when restrained fit was necessary, and where we observed a slight increase in the base polarization (10–15 mP) in competitive fitting with other competitors, we used this modified window as a restraint. The reported standard deviations and their standard deviations are the averages and standard deviations of 250–500 independent fits of simulated datasets.

The dissociation constant of the direct and competitive FP experiment was obtained by fitting the measured data with quadratic and competitive equation, respectively[63,64]. ΔG values at 295 K were calculated using the equation:

$$\Delta G = -RT * \ln(K_D) \qquad (1)$$

ΔΔG$_{av}$ values were obtained by calculating the average and the standard deviation of all obtained individual ΔΔG values (between different motifs or different proteins), excluding cases when $K_D > 300$ μM. All binding data and the obtained fits are provided as Supplementary Data File 1.

**Crystallization and structure determination**. Crystallization conditions were screened using commercially available and in-house developed kits (Qiagen, Hampton Research, Emerald Biosystems) by the sitting-drop vapor-diffusion method in 96-well MRC 2-drop plates (SWISSCI, Neuheim, Switzerland), using a Mosquito robot (TTP Labtech, Cambridge, UK) at 4 °C. The optimized condition of the crystals consisted of 19% polyethylene glycol 4000, 0.1 M cacodylate buffered at pH 5.5. For soaking, crystals were transferred to a mother–liquor solution containing (saturated, partially precipitated) 5 mM FSC and crystals were harvested after an 18-h incubation period. All crystals were flash-cooled in a cryoprotectant solution containing 20% glycerol and stored in liquid nitrogen.

X-ray diffraction data were collected at the Synchrotron Swiss Light Source (Switzerland) on the X06DA (PXIII) beamline and processed with the program XDS[65]. The crystal structure was solved by molecular replacement with a high-resolution crystal structure of 14-3-3ζ (PDB ID 2O02) using Phaser[66] and structure refinement was carried out with PHENIX[67]. TLS refinement was applied during the refinement. The crystallographic parameters and the statistics of data collection and refinement are shown in Table 1.

**Predictions of proportions of 14-3-3 isoform complexes**. We built a simple predictor tool that can be run using Microsoft Excel (Supplementary Data File 2). As an input, the predictor requires (i) the $K_D$ of binding of the phosphoprotein of interest to at least one 14-3-3 isoform, and (ii) the cellular concentrations of the seven 14-3-3 isoforms and of the phosphoprotein of interest. As an output, the predictor estimates the fraction of phosphoprotein that is engaged with each distinct 14-3-3 isoform.

From the provided $K_D$ value(s), the predictor derives all $K_D$ values for the remaining 14-3-3 isoforms, using the average relative affinity ratios described in our results.

The cellular protein concentrations required by the predictor can either be determined experimentally or fixed arbitrarily to explore hypotheses. In the present work, we used integrated protein abundance data from the PAXdb database[3]. In this database, abundance of a given protein is expressed as the ppm fraction of the number of molecules of that protein species relative to the cumulated number of all molecules of all protein species detected in the sample[68]. For instance, if the abundance of a protein species $Prot_n$ ($Ab_{Protn}$) is 1000 ppm, this means that out of a total one million ($10^6$) counted proteins, one thousand ($10^3$) correspond to the protein species of interest. Furthermore, the total intracellular protein concentration $Prot_{Tot}$ has been estimated to be around 3 mM[69]. Therefore, for any protein $Prot_n$ of interest, one can use the ppm abundance value, $Ab_{Protn}$, to roughly estimate the cellular molar concentration of that protein ($Prot_n$) using Eq. (2):

$$(Prot_n) = Ab_{Protn} \times 10^{-6} \times (Prot_{Tot}) = Ab_{Protn} \times 10^{-6} \times 3mM. \quad (2)$$

For instance, for $Ab_{Protn} = 1$ ppm,

$$(Prot_n) = 1 \times 10^{-6} \times 3\,mM = 3\,nM \quad (3)$$

for $Ab_{Protn} = 1000$ ppm,

$$(Prot_n) = 1000 \times 10^{-6} \times 3\,mM = 3\,\mu M. \quad (4)$$

While 3 mM is a reasonable estimate of the total intracellular protein concentration, one might argue that picking this particular value is an arbitrary choice, since total numbers of protein molecules may vary by up to tenfold from one cell type to another (http://book.bionumbers.org/how-many-proteins-are-in-a-cell/). However, in practice, we found that the proportions of bound 14-3-3 isoforms computed by the predictor do not significantly change for any value of $(Prot_{Tot})$ taken in the range between 1 and 10 mM.

**Reporting summary**. Further information on research design is available in the Nature Research Reporting Summary linked to this article.

## Data availability
All data supporting the findings of this study are available within the paper and its supplementary information files. Protein abundance data were taken from PAXdb (https://pax-db.org), HPV types were analyzed using the PaVE database (https://pave.niaid.nih.gov/), protein–protein interactions data were retrieved from the BioPlex 3.0 (https://bioplex.hms.harvard.edu) and kinome interactome databases (https://sec-explorer.shinyapps.io/Kinome_interactions/), and protein sequences and parameters were taken from Uniprot (https://www.uniprot.org; Uniprot IDs P31947, P63104). The refined models and the structure factor amplitudes have been deposited in the PDB with the accession codes 6ZFD (14-3-3ζ/18E6 complex; https://doi.org/10.2210/pdb6ZFD/pdb) and 6ZFG (14-3-3ζ/18E6/FSC complex; https://doi.org/10.2210/pdb6ZFG/pdb). Figures 4 and 5, as well as Supplementary Figs. 7 and 9, have associated raw data. Source data are provided with this paper.

## Code availability
The source code of the Python-based ProFit package is freely available at GitHub (https://github.com/GoglG/ProFit) and Zenodo (https://doi.org/10.5281/zenodo.4506063)[70].

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

## Acknowledgements

We would like to thank Prof. Lawrence Banks for the shared plasmids, Prof. Alexey Babakov for the provided fusicoccin preparations, Dr. Yaroslav Faletrov for fusicoccin identity verification, and Dr. Ed Huttlin for guidance on the BioPlex database. N. N. S. is grateful to the Russian Science Foundation for the grant no. 19-74-10031 covering studies of 14-3-3 proteins. Protein purification and characterization was partially done in the framework of the Program of the Russian Ministry of Science and Higher Education (K. V. T. and N. N. S.). We thank the support of the Swiss Light Source synchrotron (P. Scherrer Institute, Villigen, Switzerland) and the help of the beam-scientist at the PXIII beamline. The work was also supported by the Ligue contre le cancer (équipe labellisée 2015 to G. T.), the Agence Nationale de la Recherche (grant UBE3A ANR-18-CE92-0017 to G. T.), the French Infrastructure for Integrated Structural Biology (FRISBI), and Instruct-ERIC. G. G. was supported by the Post-doctorants en France program of the Fondation ARC.

## Author contributions

G.G. and K.V.T. contributed equally. G.G., G.T., and N.N.S. conceived the idea. G.G. purified proteins, carried out FP experiments, performed crystallographic and bioinformatics studies, analyzed the data, and edited the paper. K.V.T. cloned, purified, and characterized proteins. C.K. cloned and purified proteins. P.E. synthesized the peptides. G.T. performed data analysis, data interpretation, supervised the research, wrote and edited the paper. N.N.S. contributed to protein purification and crystallographic experiments, supervised the research, analyzed the data, wrote and edited the paper.

## Competing interests

The authors declare no competing interests.
