## [Peer Review File · Nature Communications]

REVIEWER COMMENTS

Reviewer #1 (Remarks to the Author):

The manuscript “Hierarchized phosphotarget binding by the seven human 14-3-3 isoforms” by Gogl et al. represents a multi-layered, comprehensive, and timely study that puts a number of long-standing observations in the 14-3-3 community into perspective and has also a relevance for the more general field of protein-protein interactions and protein interactomes.

The authors begin with a systematic analysis of the sequences of E6 proteins from human papillomavirus (HPV), an important and causative risk factor for the development of genital cancers. Interestingly, 31 of these proteins display phosphorylation motifs that strongly resembles the C-terminal recognition sequences for 14-3-3 binding. 14-3-3 proteins are already described as human proteins that interact with HPV E6, but for the first time, a systematic analysis of differences in binding (affinity) correlated with sequence variations in E6 measured against all 7 human 14-3-3 isoforms is presented here. These measurements delivered two important findings: a) although the E6 C-terminal sequences are very similar, their binding affinity to any individual 14-3-3 isoforms differs considerably, and b) there is a systematic order of affinity among the 14-3-3 isoforms binding to these peptides with 14-3-3gamma being the strongest and 14-3-3sigma the weakest binder.

To elucidate the structural basis of HPV-E6 C-terminal motifs binding to 14-3-3, a fusion construct between 14-3-3zeta and the HPV-18E6 C-terminus was crystallized. Crystallization of such chimaera was introduced some years ago by the main authors' group and is an elegant solution to the challenge of crystallizing a 14-3-3 binding phosphopeptide in the correct stoichiometry with 14-3-3. Moreover, these chimeras are excellent constructs to study the “three-body-problem” of 14-3-3 PPI stabilization by reducing it to a “two-body-challenge”. Satisfyingly, the corresponding crystals diffracted to a resolution of 1.9 angstrom which allowed a detailed analysis of the protein-peptide interface.

To address the molecular basis for the differences in affinity of the HPV E6 peptides, different sequence versions of the peptides were synthesized to answer the question if single residue side-chain interactions with 14-3-3 determine the binding affinity or if an overarching effect like a complementary intra-peptide preorganization (charge clamp) lowering the entropic penalty upon binding might be the reason for the observed affinity differences. Judging from the results and expressed by the authors, both mechanisms work in tandem which certainly would allow the highest degree of flexibility for regulation within the cell.

Since the binding of 14-3-3 proteins to HPV E6 is of functional consequence for these pathogenicity factors, small molecules that are able to modulate (stabilize or inhibit) these interactions could be of tremendous value for therapeutic purposes. To evaluate the accessibility of the 14-3-3/HPV E6 interface for such molecules, the authors tested the ability of the known 14-3-3 binding natural product FC-A to interfere with binding of a fluorescently labelled E6 peptide to 14-3-3 in a fluorescence polarization (FP) assay. Due to steric conflicts with the binary mode of binding of the E6 peptide, the apparent KD of its binding to 14-3-3 is reduced in the presence of FC-A. Nonetheless,

the structure of FC-A binding to the 14-3-3gamma-HPV 18E6 complex could be solved after soaking the natural product in preformed crystals. This is an important finding which shows that the 14-3-3/HPV E6 is “ligandable”.

After obtaining these important experimental results, the authors continue with what in my eyes is maybe the most valuable part of this excellent manuscript. They analyzed literature data of the observed hierarchical binding affinity of different 14-3-3 isoforms and among a wide range of 14-3-3 binding partner peptides and put this data into perspective with the results obtained in this work. In addition, the interactomes of the different 14-3-3 isoforms were compared highlighting the promiscuity of 14-3-3 isoforms like eta and zeta and the more specific interactions of sigma. Finally, and maybe even more important, the “complexomes” of 14-3-3 were predicted and discussed. The underlying thought here is to correlate the abundance of 14-3-3 in a given tissue with the affinity and quantity of their partner proteins. This information allows to predict the ratio of 14-3-3-bound and -unbound population of a 14-3-3 interaction partner with important implications for the biological overall behavior of the partner protein. This is an extremely valuable analysis and will have a deep conceptual impact in the 14-3-3 community and beyond.

I believe that this manuscript will be a landmark paper and I strongly support publication.

Sincerely,

Christian Ottmann

Reviewer #2 (Remarks to the Author):

The existence of different 14-3-3-encoding genes that are differentially expressed has time and again attracted attention but the role of the different spectrum of 14-3-3s present in different cell types remains poorly understood. The manuscript by Gogl et al. addresses this interesting problem by systematic quantitative characterisation of 14-3-3 - client protein interactions. The manuscript reflects tremendous amounts of work and reports extensive data sets and a plethora of useful information. Yet, it is not well presented using Figures that are poorly accessible due to their resolution and organisation. The flow of the text is difficult to follow. The link between Figs. 1/4 on the one hand (focus on isoforms) and 2/3 on the other hand (focus on structural determinants of binding affinity and modulation of binding by a drug) is not well developed.

The major claims are (1) a differential affinity series of the 14-3-3 proteins (homodimers) binding to the same target phosphopeptide, (2) differential affinity of the same 14-3-3 protein binding to variants of a target phosphopeptide, (3) the notion that the two differential affinity series give a unique set of 14-3-3 bound and free states depending on the respective concentrations and extent of phosphorylation of the interacting proteins. These claims are well consistent with the existing literature but I find that the authors use their claims to formulate an interesting and testable model supported by a large data set - if only the issue of the homo- versus the physiological heterodimers had been tackled.

The work could be of interest to the 14-3-3 community and a much wider field if complemented by some data and discussion addressing the heterodimer issue. The manuscript also requires extensive re-editing and much clearer presentation of the data as well as the logic of the experiments and their interpretation. Eventually the paper has the potential to influence thinking in the field.

Major points:

The introduction should include the major facts about 14-3-3 structure (dimerization helices, organisation of binding groove, determinants of binding in the binding groove, know differences between binding of isoforms to target peptides) and dimerization patterns. This background is required to follow the results and their interpretations.

Fig. 1: Does the MBP tag affect the reported affinities? Could you present selected interactions without the MBP tag? How often have these experiments been repeated to arrive at the indicated standard deviations? How relevant are these measurements given that in vivo most 14-3-3 proteins exist as heterodimers and not homodimers?

Fig. 4: This Figure is almost inaccessible due to its dense content and low resolution. If I understand the approach correctly the baits used are over-expressed 14-3-3 variants and hence homodimers. Therefore, comparison with Fig. 1 is appropriate but how these differential affinities for the 14-3-3 isoforms are physiologically relevant would definitely depend on the mix of heterodimers. Many 14-3-3 clients present multiple binding motifs and avidity effects are thus important for 14-3-3 binding to intact proteins with tertiary and quaternary structure. These aspects are not covered by the methodology used but will be relevant to the in-vivo interactome of the different 14-3-3s.

Reviewer #3 (Remarks to the Author):

The study by Gogl et al outlines the ligand specificities of the seven 14-3-3 isoforms, starting from a detailed characterisation of their interactions with the E6 oncoproteins. The authors then expand the study to discuss the specificity of the 14-3-3s in a cellular context, considering the cellular concentration of the proteins. The study is well performed and nicely put together. The integrated combination of the experiments and the computational aspects provides a novel aspect to a topic that has been studied for more than 20 years, and brings it to a level that it merits publication in a high impact journal such as Nat Com.

Minor comments:

Figure 1.

A. The authors should check the use of significant digits. For example, 4.05 ± 0.65 should be 4.1 ± 0.7 to be correct, as the other digits carries no real information. This goes for all values given in Fig 1B. Too many digits without significance makes it less easy to get a clear overview. This should be checked for all values in the manuscript.

B. Panel C is redundant as the data is shown in B. I suggest the authors to remove it, or show the gradient in B directly.

The resolution in Supplemental figure 2 is too poor to allow evaluation of the data. It is not clear if the y-axis is the same scale or not. Nevertheless, it seems like the amplitudes are different for same peptides? In some cases, it seems like the data is fitted using just a slope and a fixed amplitude? If this is the case, a varying amplitude will of course cause an issue with the fitting. Please clarify this, and provide a figure with ok resolution.

The authors should comment more on the fact that 14-3-3s both homo- and heterodimerise. While some of them prefer to homodimerize, other have preference for heterodimerisation. Please reflect on this topic in the discussion, and how taking this into account would complicate the calculations.

Point-by-point response to reviewers' comments

The authors' responses to the reviewers' comments (black font) are found below in blue font. Fragments of the revised version, used as part of our responses, are in green font.

REVIEWER COMMENTS

Reviewer #1 (Remarks to the Author):

The manuscript "Hierarchized phosphotarget binding by the seven human 14-3-3 isoforms" by Gogl et al. represents a multi-layered, comprehensive, and timely study that puts a number of long-standing observations in the 14-3-3 community into perspective and has also a relevance for the more general field of protein-protein interactions and protein interactomes.

The authors begin with a systematic analysis of the sequences of E6 proteins from human papillomavirus (HPV), an important and causative risk factor for the development of genital cancers. Interestingly, 31 of these proteins display phosphorylation motifs that strongly resembles the C-terminal recognition sequences for 14-3-3 binding. 14-3-3 proteins are already described as human proteins that interact with HPV E6, but for the first time, a systematic analysis of differences in binding (affinity) correlated with sequence variations in E6 measured against all 7 human 14-3-3 isoforms is presented here. These measurements delivered two important findings: a) although the E6 C-terminal sequences are very similar, their binding affinity to any individual 14-3-3 isoforms differs considerably, and b) there is a systematic order of affinity among the 14-3-3 isoforms binding to these peptides with 14-3-3gamma being the strongest and 14-3-3sigma the weakest binder.

To elucidate the structural basis of HPV-E6 C-terminal motifs binding to 14-3-3, a fusion construct between 14-3-3zeta and the HPV-18E6 C-terminus was crystallized. Crystallization of such chimera was introduced some years ago by the main authors' group and is an elegant solution to the challenge of crystallizing a 14-3-3 binding phosphopeptide in the correct stoichiometry with 14-3-3. Moreover, these chimeras are excellent constructs to study the "three-body-problem" of 14-3-3 PPI stabilization by reducing it to a "two-body-challenge". Satisfyingly, the corresponding crystals diffracted to a resolution of 1.9 angstrom which allowed a detailed analysis of the protein-peptide interface.

To address the molecular basis for the differences in affinity of the HPV E6 peptides, different sequence versions of the peptides were synthesized to answer the question if single residue side-chain interactions with 14-3-3 determine the binding affinity or if an overarching effect like a complementary intra-peptide preorganization (charge clamp) lowering the entropic penalty upon binding might be the reason for the observed affinity differences. Judging from the results and expressed by the authors, both mechanisms work in tandem which certainly would allow the highest degree of flexibility for regulation within the cell.

Since the binding of 14-3-3 proteins to HPV E6 is of functional consequence for these pathogenicity factors, small molecules that are able to modulate (stabilize or inhibit) these interactions could be of tremendous value for therapeutic purposes. To evaluate the accessibility of the 14-3-3/HPV E6 interface for such molecules, the authors tested the ability of the known 14-3-3 binding natural product FC-A to interfere with binding of a fluorescently labelled E6 peptide to 14-3-3 in a fluorescence polarization (FP) assay. Due to steric conflicts with the binary mode of binding of the E6 peptide, the apparent K_D of its binding to 14-3-3 is reduced in the presence of FC-A. Nonetheless, the structure of FC-A binding to the 14-3-3gamma-HPV 18E6 complex could be solved after soaking the natural product in preformed crystals. This is an important finding which shows that the 14-3-3/HPV E6 is "ligandable".

After obtaining these important experimental results, the authors continue with what in my eyes is maybe the most valuable part of this excellent manuscript. They analyzed literature data of the observed hierarchical binding affinity of different 14-3-3 isoforms and among a wide range of 14-3-3 binding partner peptides and put this data into perspective with the results obtained in this work. In addition, the interactomes of the different 14-3-3 isoforms were compared highlighting the promiscuity of 14-3-3 isoforms like eta and zeta and the more specific interactions of sigma. Finally, and maybe even more important, the “complexomes” of 14-3-3 were predicted and discussed. The underlying thought here is to correlate the abundance of 14-3-3 in a given tissue with the affinity and quantity of their partner proteins. This information allows to predict the ratio of 14-3-3-bound and -unbound population of a 14-3-3 interaction partner with important implications for the biological overall behavior of the partner protein. This is an extremely valuable analysis and will have a deep conceptual impact in the 14-3-3 community and beyond.

I believe that this manuscript will be a landmark paper and I strongly support publication.

Sincerely,

Christian Ottmann

Response: We are deeply grateful to Prof. Ottmann for his remarkable review, which faithfully and comprehensively summed up all our findings, interpretations and conclusions, emphasizing those of particular interest to the 14-3-3 research community and beyond. The fact, that Prof. Ottmann so clearly captured the essence of all the messages we wished to put forward in the manuscript, makes us confident that the data were, overall, properly presented in the first version of our manuscript.

Reviewer #2 (Remarks to the Author):

Rev#2-1. The existence of different 14-3-3-encoding genes that are differentially expressed has time and again attracted attention but the role of the different spectrum of 14-3-3s present in different cell types remains poorly understood. The manuscript by Gogl et al. addresses this interesting problem by systematic quantitative characterisation of 14-3-3 - client protein interactions. The manuscript reflects tremendous amounts of work and reports extensive data sets and a plethora of useful information.

Response: We would like to thank reviewer #2 for such a high mark to the work done in our study.

Rev#2-2. Yet, it is not well presented using Figures that are poorly accessible due to their resolution and organisation.

Response: We fully agree with this. Actually, we now realized that we had a problem with the uploading of the figures at the submission, which resulted in figures of very low resolution. We sincerely apologize for that. This new submission provides figures of optimal resolution.

Rev#2-3. The flow of the text is difficult to follow. The link between Figs. 1/4 on the one hand (focus on isoforms) and 2/3 on the other hand (focus on structural determinants of binding affinity and modulation of binding by a drug) is not well developed.

Response: We thankfully acknowledge the comment of reviewer #2. However, based on reviewer #1 and #3, which reported that this is "an excellent manuscript" and that our story is "well performed and

nicely put together”, we chose not to alter dramatically the plot and the overall organization of the paper. Otherwise, we would run the risk of disappointing both reviewers #1 and #3 without satisfying reviewer #2. Nevertheless, we did address all the precise points raised by reviewer #2, as well as reviewer #3 (see below).

Rev#2-4. The major claims are (1) a differential affinity series of the 14-3-3 proteins (homodimers) binding to the same target phosphopeptide, (2) differential affinity of the same 14-3-3 protein binding to variants of a target phosphopeptide, (3) the notion that the two differential affinity series give a unique set of 14-3-3 bound and free states depending on the respective concentrations and extent of phosphorylation of the interacting proteins. These claims are well consistent with the existing literature but I find that the authors use their claims to formulate an interesting and testable model supported by a large data set - if only the issue of the homo- versus the physiological heterodimers had been tackled. The work could be of interest to the 14-3-3 community and a much wider field if complemented by some data and discussion addressing the heterodimer issue.

Response: We are grateful to the reviewer for the accurate reformulation of the main claims of our study and the note that its results are very well consistent with the literature and proteomics data. We acknowledge that the heterodimers of 14-3-3 proteins were not discussed. We have now addressed this in the revised version. Please also see the detailed response addressing the heterodimer issue below (response to Rev#2-8).

Rev#2-5. The manuscript also requires extensive re-editing and much clearer presentation of the data as well as the logic of the experiments and their interpretation. Eventually the paper has the potential to influence thinking in the field.

Response: please see our previous response to Rev#2-3.

Major points:

Rev#2-6. The introduction should include the major facts about 14-3-3 structure (dimerization helices, organisation of binding groove, determinants of binding in the binding groove, know differences between binding of isoforms to target peptides) and dimerization patterns. This background is required to follow the results are their interpretations.

Response: We added such a paragraph to the introduction (see below in green, as marked also in the revised manuscript file). We also cited appropriate reviews for any reader interested in further details.

« 14-3-3 proteins have a highly conserved dimeric all-helical structure 1, 4. Each monomer is formed by a bundle of nine antiparallel helices: the N-terminal $\alpha 1$ - $\alpha 4$ helices comprise a dimerization zone and a bottom of the cup-shape dimer, whose walls are built by the C-terminal α -helices 5. Each monomer features a well-conserved amphipathic groove and much less conserved convex solvent-exposed face and the disordered C-terminal tail 1, 4. 14-3-3 proteins can form homodimers or heterodimers comprising two different isoforms 4, 6. According to various observations in vitro and in cells, 14-3-3 σ preferentially homodimerizes, 14-3-3 ϵ preferentially heterodimerizes (with any isoform except 14-3-3 σ), whereas other isoforms tend to indifferently homodimerize or heterodimerize 4. Heterodimerization preferences can be explained, at least in part, by the number of intermolecular salt bridges that can occur at the dimer interface 4. However, to our knowledge, no structure of any 14-3-3 heterodimer has been released to date, and a comprehensive study of homo- and heterodimerization affinity and/or

kinetic constants of all isoforms is still awaited. The cellular proportions of homo- and heterodimers are likely to vary depending on numerous factors such as the cellular concentrations of each isoform, their turnover rates, localization and post-translational modifications, which in turn will all vary depending on cell type and cellular status.

14-3-3 isoforms all have the ability to bind phosphopeptides 7, 8. Each monomer can bind one phosphopeptide via its amphipathic groove. Consequently, a 14-3-3 dimer can bind two phosphosites simultaneously. Those can originate from two different regions of the same protein, or from two different proteins. »

Rev#2-7. Fig. 1: Does the MBP tag affect the reported affinities? Could you present selected interactions without the MPB tag? How often have these experiments been repeated to arrive at the indicated standard deviations?

Response: We did not observe any evidence that would indicate that the MPB tag affects the resulting affinities. Before completing our panel of homodimeric 14-3-3, we made preliminary tests using 3 untagged 14-3-3 proteins (beta, tau, epsilon) devoid of the flexible C-terminal tails. In this experiment, we observed the same order of affinity hierarchy between the different phosphorylated E6 partners. This indicated that neither the MBP tag nor the flexible C-terminal tails affect the general affinity profiles that we unearth, which reinforces the finding one more time. We added these data in a new Supplementary Fig. 2c and also provided an additional statement on these observations in the Results section :

« Of note, the presence of the MBP tag did not affect the relative affinity differences observed for the 14-3-3 isoforms since the untagged selected 14-3-3 isoforms obeyed the same trend from the strongest to the weakest (tau/beta and epsilon) (Supplementary Fig. 2c) observed for the MBP-tagged variants (Fig. 1b and c). The unfused 14-3-3 variants also preserved preferences for the four selected HPVE6 PBM (Supplementary Fig. 2c) described in more detail for the full-length MBP-tagged 14-3-3 isoforms (Fig. 1). »

In addition, in our final analysis, when we compare our results with results from past publications, we observed the same affinity trends, although those experiments were done using different constructs and different methods.

All FP experiments were done in triplicates. Data were fitted with ProFit that utilizes a Monte Carlo approach to take into account experimental variability. The reported affinities and their standard deviations are the averages or standard deviations of 250-500 independent fits of simulated datasets. The fitting procedure was described in Simon et al. FEBS, 2019 and the source code is freely available on Github <https://github.com/GogIG/ProFit>. This is now reflected in the Code availability statement. Moreover, as part of the revision, we provide the raw ProFit input files that are sufficient to reproduce all presented fits. The .ZIP folder is now provided as a Source data file.

Rev#2-8. How relevant are these measurements given that in vivo most 14-3-3 proteins exist as heterodimers and not homodimers?

Response: We now address this issue in the discussion (see below in green font). Anyway, one important warning to readers, now added in that discussion paragraph, is that the proposed predictor is a rather "unrefined" tool mainly intended to stimulate thinking and test hypotheses. In our opinion,

heterodimerization and multivalency are only two instances of many more molecular complexities that we may still be missing in our representations of what a cellular environment really is.

« While 14-3-3 proteins predominantly exist as dimers, the predictor deals with concentrations of 14-3-3 isoform monomers. The calculation assumes that the affinity of each monomer molecule towards a single phospholigand corresponds to the affinity measured for homodimers and is not influenced by the nature of the neighbor monomer, be it the same isoform (homodimeric species) or of another isoform (heterodimeric species). This assumption is plausible, considering the very high conservation of the amphipathic grooves of 14-3-3 proteins (Supplementary Fig. 8), responsible for ligand binding and facing each other in the dimeric structures. Anyhow, our predictor should be mainly intended as a "rough trend estimator" to stimulate thinking and explore hypotheses, rather than an accurate descriptor of the actual precise proportions of 14-3-3 complexes in cells. »

This being done, we would however like to make a few remarks, as concerns heterodimerization and the credit this concept receives in the field.

1) There are hundreds of structures of 14-3-3 proteins available, yet, to the best of our knowledge, they only show homodimers, without exception. Structure determination of a 14-3-3 heterodimer complexed with different phospholigands has not been possible so far, therefore, our understanding of the heterodimer/phosphopeptide interactions are limited. In addition, the isolation of highly stable and homogenous heterodimers seems to be experimentally challenging.

2) Similarly, we do not know of any published work which would have systematically addressed all homo- and hetero-dimerization pairwise affinities of the seven human isoforms. Such data seem to be missing in the field, considering the importance given by the field to homodimerization and heterodimerization trends. If we want to apprehend better the *in vivo* impact of heterodimers, we need studies that (i) measure heterodimerization affinities and (ii) measure the affinities of preformed heterodimers to third-party phosphotargets, which might be modulated and further complicated by multimeric ligands.

3) In any case, we would like to notice that the proportion of 14-3-3 heterodimers *in vivo* might turn out to be more modest than is generally thought. Yang X et al (Proc Natl Acad Sci USA 2006;103:17237–42) have noticed that 14-3-3 homodimer-heterodimer exchange is a slow process that requires several hours to be completed («We also studied the formation of heterodimers between four different 14-3-3 isoforms. Different isoforms were mixed and left overnight before measurements were taken. Shorter incubation periods, for instance, of 1 h, did not allow the subunit exchange to proceed to completion.», p. 17238, right). Ghorbani et al (Amino Acids (2016) 48:1221–1229) and Tugaeva et al (FEBS J. 2020 Sep;287(18):3944-3966) also used the overnight incubation to achieve efficient formation of 14-3-3 heterodimers.

As 14-3-3 proteins are highly abundant, they are most likely produced by polysomes, where several ribosomes are simultaneously producing proteins from the same molecule of mRNA. In such a situation, 14-3-3 monomers are likely to immediately homodimerize with atomically proximal monomers produced from the same mRNA, found immediately at hand just after synthesis (co-translational folding). Once formed, such homodimers will take several hours before reaching a fully equilibrated exchange with all other homodimers. This may even be slowed down by the fact that, in the meantime many freshly produced homodimers may engage with third-party phosphotargets, further reducing their capacity to break off for exchanging with other homodimers. Such phenomena may create a kinetic bias favoring homodimers of all isoforms, including 14-3-3 epsilon (which is known to thermodynamically prefer heterodimerization *in vitro*). An important parameter will be the turnover (translation-

degradation) rates of 14-3-3 molecules. Even if their turnover rate is in the order of days, homodimers might dominate the scene during some hours after synthesis and will only then gradually transmute into various mixtures of dimers according to their thermodynamic preferences. We should also take into account that the elegant founding experiment of Chaudhry et al (BBRC 2003 300:679–685), which demonstrated formation of heterodimers in cells, was biased to the formation of heterodimers since the tagged 14-3-3 proteins utilized for heterodimer immunoprecipitation was strongly overexpressed as compared to the endogenous untagged 14-3-3 isoforms. These problems were also discussed in the pioneer study by Jones et al FEBS Lett 1995 368: 55-58, in which only a very inefficient heterodimerization of 14-3-3 proteins occurred in vivo, and only when both tagged and untagged 14-3-3 versions were transfected (and overexpressed) in the cells. The overexpression of the fused tagged 14-3-3 isoforms resulted in almost no heterodimerization.

In fact, Chaudhry et al (BBRC 2003 300:679–685) also performed experiments demonstrating heterodimerization of yeast 14-3-3 proteins in the absence of any overexpression. However, there are only two 14-3-3 isoforms in yeast instead of seven in human, and their turnover rate and heterodimerization propensity might be seriously different, so it is unclear to which point the observation in yeast can be extended to human.

To sum up, we would like to mention that our in vitro interactomics data obtained for the human 14-3-3 homodimers (and supported by data from literature obtained using homodimers as well) are in excellent agreement with the proteome-wide Bioplex data obtained in cells (summarized in Fig. 4). This may serve as an indirect indication that the role of 14-3-3 heterodimers is not so critical on the proteome-wide scale, but this warrants further, systematic investigation.

Rev#2-9. Fig. 4: This Figure is almost inaccessible due to its dense content and low resolution.

Response: Resolution has now been fixed. We also acknowledge that our former Fig. 4 was definitely "very busy" with a lot of important information. Indeed, this figure was dealing with two different subparts of our story, as explained in Prof. Ottmann's report. While the first part of the figure dealt with the "affinity-based interactome" of 14-3-3 proteins, the second part dealt with their "cellular complexomes". Actually, these two parts corresponded to two separate result sections.

We have now presented these data as two figures: new Fig. 4 for the interactome and new Fig. 5 for the complexome; each related to their own separate result sections.

In the new Fig. 4 we now added a panel (i) showing the correlation between hierarchical affinity ranking and sequence divergences, formerly shown in supplemental Fig. 9. We feel this information is important as it provides a link between affinity variations and structural features imbedded in the sequences.

As concerns Fig. 4, we also performed additional modifications, for the following reasons. During the reviewing phase of the paper, we managed to contact Dr. Ed Hutlin, the Harvard scientist in charge of the Bioplex project. Thanks to this discussion, we realized that our initial interpretation of the "baits" and the "preys" in their database listings was not fully correct. Dr. Hutlin provided us with all the necessary information to extract the lists of baits and preys correctly. We have re-done the plots correspondingly. Importantly, these changes do not alter the global outcome of our findings. We still find an excellent correlation between the number of interactions engaged by each 14-3-3 isoform according to the database, and the relative affinity of the each 14-3-3 isoform for phosphotargets.

In addition, Dr. Hutlin also instructed us on how to extract from their database the so-called PSM values (Peptide-Spectra Matches), which correspond to quantitative relative amounts of prey proteins as determined by mass spectrometry. By carefully analyzing these data as concerns the 14-3-3-binding preys, we observed a good correlation between our affinity trends and the relative amounts of enriched preys. These new data have now been added as a new panel of Fig. 4 (h) in the revised manuscript. Therefore, this demonstrates that the Bioplex database is of sufficiently high quality not only to count all the distinct 14-3-3-target pairwise interactions (as we had already done in our previous version), but also to extract PSM-based information about relative binding strengths of the seven 14-3-3 isoforms, which turn out to be, in large part, correlated with the 14-3-3/phosphotarget relative affinity scale that we have described. This is an additional finding which further reinforces our claims in the manuscript. The main text has been edited accordingly.

« Furthermore, we observed a remarkable linear correlation ($R^2 = 0.91$) between the numbers of binders detected by the Bioplex project 29 for each 14-3-3 isoform, and their relative affinity ($\Delta\Delta G_{av}$) as compared to the strongest phosphopeptide-binding isoform, 14-3-3 γ (Fig. 4g).

In the AP-MS experiments, interaction partners (and 14-3-3 proteins in particular) can be either “baits” or “preys”. Baits are recombinantly expressed in the cells using the same promoter, which should ensure a relatively even expression for all 14-3-3 isoforms. By contrast, the preys are proteins naturally expressed by the cells, so that the distinct 14-3-3 preys should be present in different amounts, depending on their intrinsic levels of expression in the host cells. In Bioplex, six out of seven 14-3-3 isoforms (with the exception of ϵ) were among the tested recombinant bait proteins. This allowed us to distinguish, among the 14-3-3 binders, those identified as baits retaining 14-3-3 preys, from those identified as preys retained by 14-3-3 baits. In both situations, the linear correlation of the numbers of 14-3-3 binders with the relative phosphopeptide-binding affinity scale of 14-3-3 proteins remained very strong ($R^2 = 0.96$ and $R^2 = 0.90$, respectively) (Supplementary Fig 7a).

In further support of these interrelations, the number of 14-3-3 prey-binding baits also indicated correlation with the affinity trend of 14-3-3 isoforms when using data from a recent independent study (https://sec-explorer.shinyapps.io/Kinome_interactions/ and 49) that used AP-MS to uncover the interactions of more than 300 protein kinases ($R^2 = 0.64$) (Supplementary Fig. 7a).

The Bioplex database 29 also contains all peptide-spectrum matches (PSM) values for all the preys retained by each and every bait. PSM values bear information about the enrichment of prey proteins on resins carrying particular baits. The higher the PSM values measured for a given prey precipitated by a particular bait, the more enriched the prey, and, therefore, the stronger the affinity of the corresponding bait-prey complex. We retrieved and summed up the PSM values of the 114 preys captured in the Bioplex experiment 29 by five different 14-3-3 isoforms. These values show good agreement ($R^2 = 0.66$) with the affinity trends of the different 14-3-3 isoforms (Fig. 4h). Even more remarkably, when those 114 individual preys are ranked from their highest to lowest PSM values relative to 14-3-3 γ (Supplementary Fig. 7b), one observes the same bi-directional decreasing intensity pattern as seen in our experiments (Fig. 1b) as well as in the low-throughput data from literature (Fig. 4a).

Altogether, these analyses indicate that both the numbers and the PSM enrichment values of partner proteins of 14-3-3 isoforms detected by proteome-wide interactomic studies are remarkably correlated with their relative phosphopeptide-binding affinity trends. »

Rev#2-10. If I understand the approach correctly the baits used are over-expressed 14-3-3 variants and hence homodimers. Therefore, comparison with Fig. 1 is appropriate but how these differential affinities for the 14-3-3 isoforms are physiologically relevant would definitely depend on the mix of heterodimers. Many 14-3-3 clients present multiple binding motifs and avidity effects are thus important for 14-3-3 binding to intact proteins with tertiary and quaternary structure. These aspects are not covered by the methodology used but will be relevant to the in-vivo interactome of the different 14-3-3s.

Response: Please see our previous detailed response to Rev#2-8.

Reviewer #3 (Remarks to the Author):

Rev#3-1. The study by Gogl et al outlines the ligand specificities of the seven 14-3-3 isoforms, starting from a detailed characterisation of their interactions with the E6 oncoproteins. The authors then expand the study to discuss the specificity of the 14-3-3s in a cellular context, considering the cellular concentration of the proteins. The study is well performed and nicely put together. The integrated combination of the experiments and the computational aspects provides a novel aspect to a topic that has been studied for more than 20 years, and brings it to a level that it merits publication in a high impact journal such as Nat Com.

Response: We are happy to receive this kind and positive feedback of the reviewer.

Minor comments:

Figure 1.

Rev#3-2. A. The authors should check the use of significant digits. For example, 4.05 ± 0.65 should be 4.1 ± 0.7 to be correct, as the other digits carries no real information. This goes for all values given in Fig 1B. Too many digits without significance makes it less easy to get a clear overview. This should be checked for all values in the manuscript.

Response: This has been done, also for Table 1 with crystallographic statistics.

Rev#3-3. B. Panel C is redundant as the data is shown in B. I suggest the authors to remove it, or show the gradient in B directly.

Response: We have merged panel B and C as recommended.

Rev#3-4. The resolution in Supplemental figure 2 is too poor to allow evaluation of the data. It is not clear if the y-axis is the same scale or not. Nevertheless, it seems like the amplitudes are different for same peptides? In some cases, it seems like the data is fitted using just a slope and a fixed amplitude? If this is the case, a varying amplitude will of course cause an issue with the fitting. Please clarify this, and provide a figure with ok resolution.

Response: Resolution has now been fixed. All FP-data were fitted with ProFit, our recently developed fitting tool. This program takes into account both the direct and the competitive data during fitting and either forces the amplitudes of the direct titration in the competitive fitting, or it leaves the amplitudes of the competitive fit to move freely and evaluates the discrepancies of the amplitudes after fitting. As an output, it always produces pairs of fits on the same scale. Therefore, all measurements performed

with the same 14-3-3 protein are shown on the same scale. However, the amplitude of the measurements can differ between experiments done with different 14-3-3 proteins, or in different experimental conditions. In cases where the studied concentration range of the competitive titration was not sufficient for complete competition, the dissociation constant was determined using a fixed amplitude, derived from the direct titration. In this case, the sole unknown parameter during the fit is the dissociation constant of the competitor, and although the resulting fit looks like a simple slope, the deduced parameter is still rather overdetermined by our multiple datapoints.

In the case of all 14-3-3 proteins, we observed a slight increase in the base polarization during competitive fitting. This parameter was independent of the nature of the studied competitor. When we had to use a fixed amplitude window during fitting (e.g., for very weak interactions), we took into account this change in the polarization window as well. This is a rather common phenomena, which we discuss in detail in Figure 2C of Simon et al. FEBS J 287, 2834-2846 (2020).

We hope that the high-resolution version of the figure (Supplementary Fig. 2) will elucidate all questions regarding data handling but we also provide more details about the fitting procedure in the revised version :

« All FP experiments were done in triplicates. Analysis of FP experiments were carried out using ProFit, an in-house developed, Python-based fitting program 64. ProFit utilizes a Monte Carlo approach to take into account experimental variability. It generates hundreds of simulated datasets, based on the experimental data variance and fits direct and competitive measurements in pairs. The experimental polarization window is first determined in the direct experiment, then this is either used as a fixed restraint in competitive fits or as a reference value to validate the result of unrestrained fits. In cases when restrained fit was necessary, and where we observed a slight increase in the base polarization (10-15 mP) in competitive fitting with other competitors, we used this modified window as a restraint. The reported affinities and their standard deviations are the averages or standard deviations of 250-500 independent fits of simulated datasets. »

Rev#3-5. The authors should comment more on the fact that 14-3-3s both homo- and heterodimerise. While some of them prefer to homodimerize, other have preference for heterodimerisation. Please reflect on this topic in the discussion, and how taking this into account would complicate the calculations.

Response: Please see our detailed response to Rev#2-8.

REVIEWERS' COMMENTS

Reviewer #2 (Remarks to the Author):

The authors have addressed all concerns diligently and with great effect for the manuscript. It improved tremendously and will be an important resource for this field and beyond. The conceptual implications reflected in the manuscript and the review correspondence will stimulate the discussion of 14-3-3 proteins and protein-protein interactions at large, including use of datasets available in databases.